# Small body size is associated with increased evolutionary lability of wing skeleton proportions in birds

Andrew Orkney [1] ✉ & Brandon P. Hedrick [1] ✉

Birds are represented by 11,000 species and a great variety of body masses. Modular organisation of trait evolution across birds has facilitated simultaneous adaptation of different body regions to divergent ecological requirements. However, the role modularity has played in avian body size evolution, especially small-bodied, rapidly evolving and diverse avian subclades, such as hummingbirds and songbirds, is unknown. Modularity is influenced by the intersection of biomechanical restrictions, adaptation, and developmental controls, making it difficult to uncover the contributions of single factors such as body mass to skeletal organisation. We develop a novel framework to decompose this complexity, assessing factors underlying the modularity of skeletal proportions in fore-limb propelled birds distributed across a range of body masses. We demonstrate that differences in body size across birds triggers a modular reorganisation of flight apparatus proportions consistent with biomechanical expectations. We suggest weakened integration within the wing facilitates radiation in small birds. Our framework is generalisable to other groups and has the capacity to untangle the multi-layered complexity intrinsic to modular evolution.

Skeletal evolution across vertebrates is often modular[1] whereby traits covary more strongly with other traits belonging to the same, rather than separate, developmental or functional complexes[2–5]. For example, morphological variation in mammalian vertebrae clusters into functional and developmental regions along the spinal column[6,7] and morphological variety of avian cranial bones exhibit stronger evolutionary correlations within tissues of the cranial neural crest and mesodermal lineages[8]. Intraspecific trait covariance is likely to emerge as a consequence of growth and development[9], while trait covariance between species – known as evolutionary modularity – is also shaped by macroevolutionary divergence between lineages[4]. Modular organisation may establish limits on phenotypic variety, available pathways, and predisposed trajectories that lineages take through evolutionary landscapes[10–12]. Both dis-aggregation[13–15] and consolidation of modular structures[16] may underlie mosaic acquisitions of novel innovations[17], such as new styles of locomotion[18], respiration[6], and reproduction[19]. Furthermore, cascades of synchronous changes across functional modules may increase evolutionary rates[20,21]. Prevailing schemes of evolutionary modularity are therefore likely to reflect a highly complex combination of the underlying proclivity for directions of evolutionary change, ancestral histories, functional associations, and developmental trajectories[22]. This complexity poses a challenge to researchers seeking to understand the salient factors that induce or inhibit fundamental shifts in modular organisation and the role they may have played in essential mechanisms of major evolutionary change. Untangling the complexity underlying interspecific modularity therefore has the potential to greatly improve insight into the accumulation of biodiversity and disparity, and the processes of ecological adaptation underlying the history of life on Earth.

Body mass evolution is an example of a major mode of evolutionary change and ecological adaptation, which has a complex multifaceted relationship with factors integral to modular evolution, such

[1]College of Veterinary Medicine, Department of Biomedical Sciences, Cornell University, 930 Campus Rd, Ithaca, NY 14853, USA.
✉e-mail: aco58@cornell.edu; bph54@cornell.edu

as organismal biomechanics, growth and development[23–25]; and changes in skeletal proportions commensurate with body mass (interspecific allometry) are known to covary with trait integration across the skeleton[26,27]. Larger body masses can impose strong stabilising selection because high mechanical stresses demand greater consolidation of heavily-loaded structures and coordinated evolution between supporting elements[28], inhibiting their dis-aggregation into separate modules, and potentially limiting evolvability[12,29]. Thus, reductions in body mass may alleviate these constraints and permit the origin of new modules. Macroevolutionary changes in body mass are therefore likely to play a fundamental role in modular reorganisation[30,31]. Intuitive relationships between body mass, biomechanics and modular organisation make body mass an ideal candidate variable to investigate modular reorganisation and to determine whether it is possible to disentangle complex multifaceted controls on trait integration and isolate single salient narratives.

Birds are the most diverse extant amniote clade, consisting of over 11,000 species[32] the sole living representatives of Dinosauria, and are typified by a great variety of ecological niche specialisations. Previous research has shown that modular organisation differs subtly between Telluraves (the land-bird clade) and the paraphyletic non-Telluraves grade[33], which tend to have larger body masses. This invites the possibility that there is a complex intersection between body mass, modular organisation, and the gross dynamics of evolutionary disparification across birds. However, the degree of evolutionary lability of modular organisation across birds– and its potential drivers– have yet to be systematically investigated, despite mosaic flexibility in the modular evolution of the proportions of the the avian locomotory system being an established explanation for avian evolvability[18,34]. This makes birds an excellent model system to explore the relationship between body mass and modular evolution, as they constitute a hyper-diverse radiation of ecologically and morphologically disparate species distributed over 4 orders of magnitude of body mass[35]. We also reflect that the evolutionary history of birds has been critically defined by the exploration of small sizes inaccessible to other dinosaur lineages,[25,36] and that the smallest bodied avian subclades underwent rapid Cenozoic diversifications (e.g. ref. [37]), although the history of body size evolution through the Mesozoic remains fragmentary[38]. This raises the possibility that there may be a novel intersection in birds between body mass and intrinsic controls on evolutionary dynamics such as modular organisation.

Forelimb-propelled birds are especially relevant to studies of evolutionary modularity because of the intense mechanical constraints required by flight[39,40]. We observe that the primary feathers occupy a greater portion of the total avian wingspan in small birds[41], while individual skeletal proportions scale with isometry[42]. We hypothesise that the accessory mechanical function of ancillary structures with lower stiffness than the wing skeleton, such as feather remiges and supporting musculature, are able to accommodate a portion of the consolidating structural role that might be more incumbent on the skeleton in massive birds experiencing greater mechanical stresses. We therefore have a clear reason to anticipate weaker integration of wing skeleton proportions within small birds. Furthermore, Nudds et al.[41] suggest that longer primary remiges facilitate differential joint placement along the avian wing and access to novel wing stroke kinematics in smaller birds, leading us to expect that a reduction in integration within the avian wing skeleton may permit the exploration of a broader remit of flight-styles.

Here, we develop a novel framework to untangle the complexity inherent to modularity and explore the impact of body mass on the modular organisation of skeletal proportion evolution in birds.

The assessment of variation in evolutionary modularity across birds in the context of body mass is a natural experiment within which to explore the significance and role of evolutionary modularity within a highly successful vertebrate radiation. We show that the relative sizes

of individual wing bones evolve more independently of one another in small birds, consistent with our expectation that greater mechanical stresses in larger birds should cause consolidating evolutionary integration. Finally, this represents a case-study for the generalisation of our approaches to dissect modularity in other clades and questions. In particular, we advocate the further exploration of 3D skeletal morphology, remige proportions and soft tissue anatomy in bird wings.

## Results

### Overview of variance in avian skeletal proportions, body mass, and their intersection

The phylogenetic signal of the distribution of body mass across birds indicates that closely related birds are more likely to have similar masses than under a Brownian motion model ($K = 1.53, \lambda = 0.92$). This could indicate that the effects of evolutionary non-independence are not fully represented by phylogenetic comparative methods that assume Brownian evolution. This observation makes ancillary analyses necessary to verify acceptable type-1 error rates (see method 2). We note Passeriformes and Apodiformes are clearly distinguished by consistently low body masses, but that small birds occur widely throughout the phylogeny (e.g. *Todus, Alcedo, Indicator, Columbina*). The largest birds are also dispersed across the avian tree, with little preference for a single subclade (e.g. *Ardeotis, Vultur, Grus, Spheniscus, Leptoptilos, Crax*). Interspecific variance in allometrically-corrected skeletal element sizes typically increases with body mass (Fig. 1b). An ancestral state reconstruction (Fig. 1a) indicates a moderate to large body mass (≈400 g) at the root of crown birds, and that the smaller body masses typical of Apodiformes, Passeriformes, and Coraciimorphae are likely to be derived. The relative sizes of the more distal skeletal elements within the head and leg (e.g. mandible, tarsometatarsus) are the most variable. However, there is no clear evidence that the relative sizes of more distal elements within the wing are more variable than proximal elements. Furthermore, the relative sizes of skeletal elements within the trunk exhibit a disorganised increase in variance with body mass. We note that those skeletal elements which did not conform to our expectations are generally major components of the avian flight apparatus, a point we return to in the discussion.

### Skeletal evolution in small birds is characterised by a reorganisation of evolutionary modules

Our analyses of within-module integration (see method 1) show that there is a substantial increase in evolutionary integration of skeletal element sizes within the wing with increasing body mass (Fig. 2a). This means that evolutionary changes in the size of any single wing bone are more likely to be accompanied by commensurate changes in the sizes of other wing bones in large birds, and that wing bone proportions are more likely to evolve independently of one another in small birds. By contrast, evolutionary integration of skeletal element sizes within the trunk declines with increasing body mass (Fig. 2b), and there are no clear trends within the head or leg (Fig. 2c, d). Ordinary least squares fits (Fig. 3a) and illustrative micro-plots (Fig. 3b) indicate an overall decrease in cross-module integration between the wing and trunk in larger birds. Concerted evolutionary changes in small birds therefore tend to have a different anatomical distribution in contrast to their larger-bodied relatives, with wings evolving independently of the trunk in large birds, but becoming more likely to evolve together as a single unified system in small birds.

### The divergent evolution of distinctive avian taxa contributes to some aspects of modular reorganisation, but there is also widespread concordance across avian phylogeny

Breusch–Pagan tests for unequal scatter (see method 2) reveal a significant decrease in scatter in the relationship between humerus and carpometacarpus size $D_m$ (Divergence from major axis of covariation)

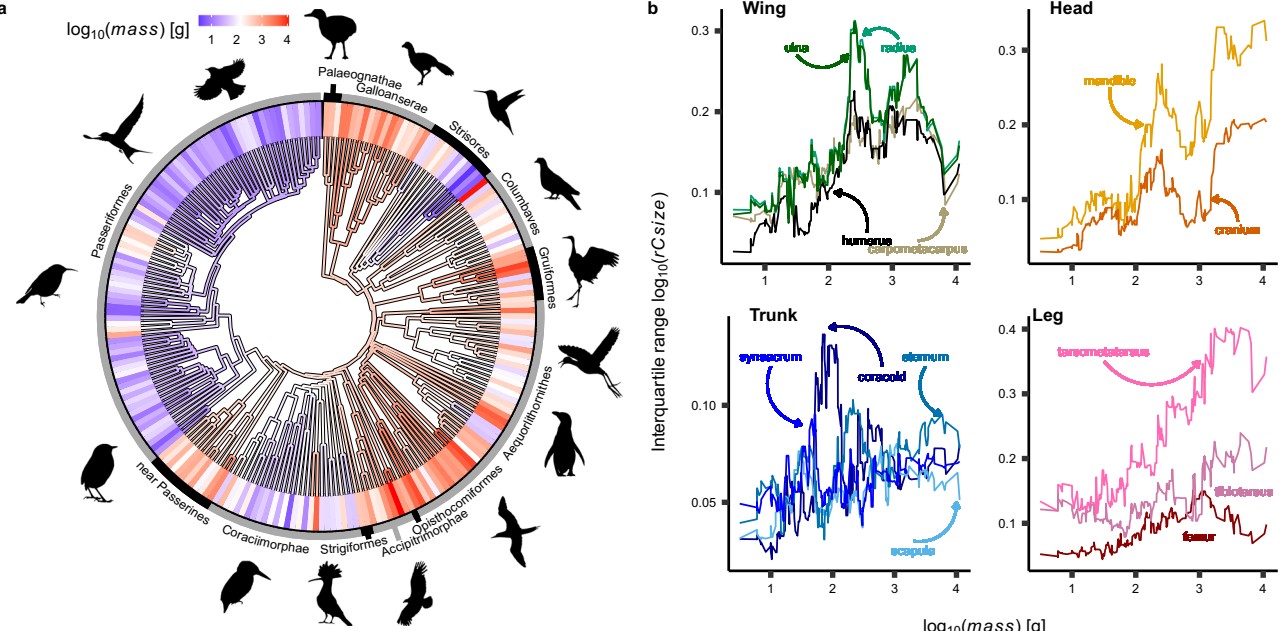

**Fig. 1 | Avian body mass and variance structure. a** Body mass variety across birds and ancestral state reconstruction (node estimates under a Brownian walk). Blomberg's *K* (1.53) and Pagel's *λ* (0.92) values indicate that phylogenetic autocorrelation is more phylogenetically clustered than expected under Brownian motion. **b** Variance structure, as represented by a running inter-quartile range of allometrically-corrected centroid sizes (log$_{10}$(*rCsize*); bin width = 30), across the avian body plan. Bones are grouped within known evolutionary modules (wing, head, trunk, leg). Skeletal proportions become more variable in larger birds, especially in the distal leg. Variance structure increases with body mass in the wing, but there is no proximo-distal gradient. Sample size *n* = 228 species. Silhouettes were sourced from phylopic.org. *Acanthorhynchus tenuirostris* (Michael Scroggie),

*Alcedo atthis* (Margot Michaud), *Archilochus colubris* (Andy Wilson), *Grus canadensis* (Sharon Wegner-Larsen; reflected horizontally), *Hirundo rustica* (Matt Wilkins), *Jacana spinosa* (Burton Robert; reflected horizontally), *Spheniscus humboldti* (Juan Carlos Jerí), *Turdus pilaris* (Sharon Wegner-Larsen), *Upupa epops* (Ferran Sayol) and *Vultur gryphus* (Ferran Sayol) were procured under a CC0 1.0 Public Domain license. *Columba palumbus* (Anthony Caravaggi), *Sula sula* (Larry Loos (photography) and Michael Keesey (vectorisation)) and *Xenicus gilviventris* (Wynston Cooper (photo) and Albertonykus (silhouette)), were procured under a CC BY-NC-SA 3.0 DEED license. *Crax alector* (J. N. Wiegers; reflected horizontally) and *Crypturellus variegatus* (J. N. Wiegers) were procured under a CC BY 4.0 DEED license.

with increasing body mass (real statistic distribution exhibits minimal overlap with null distribution), indicating increasing within-wing integration at higher body masses. Divergent evolution between Passeriformes (e.g. *Malurus*) and Apodiformes (e.g. *Archilocus, Chaetura, Streptoprocne*) contributes to this pattern, but there is also a wide divergence between lineages within Passeriformes (e.g. *Hirundo*) (Fig. 4a, b). *Puffinus, Pterodroma,* and *Fulmarus,* all pelagic soaring birds belonging to Procellariiformes, are notable outliers. These birds have relatively high body masses (705 g, 810 g, and 1000 g, respectively) but they are not predisposed to be outliers in any single direction.

There is no indication that integration between the cranium and mandible changes with body mass (Fig. 4c; real and null distributions overlap). This observation confirms that our re-sampling scheme is robust to type-1 errors that might be induced by non-Brownian phylogenetic structure and skewness in log$_{10}$(*mass*). *Archilocus* and *Probosciger*, which have unusually large and small mandibles respectively, are prominent outliers.

Scapula size-sternum size $D_m$ exhibits a sub-significant increase in larger birds and the distribution of the real test statistic clearly tends to be biased higher than the null (Fig. 4d), but the separation is not as explicit as it is for evolutionary covariance between the carpometacarpus and humerus. We observe that *Gavia* and *Rollandia*, which are nested within subclades of Aequorlithornithes, both have very large sterna, with deep keels, relative to the size of their scapulae. Positive outliers in this space, such as *Ortalis* and *Ixobrychus*, have unusually small sterna. Outliers are widely distributed across phylogeny, and a sub-significant trend for weakening integration within the trunk is therefore not driven by the divergent evolution of any particular avian subclades.

Integration between carpometacarpus size-sternum size $D_m$ is characterised by a significant increase in scatter in larger birds (Fig. 4e, real distribution is clearly biased higher than the null distribution). This suggests that integration between the trunk and wing is higher among smaller birds than large birds. Outliers within the carpometacarpus-sternum integration space include *Cathartes* and *Rhyncops*, which both have relatively large carpometacarpi compared to their sterna, while *Uria, Spheniscus, Pelecanoides* and *Crypturellus* all have relatively small carpometacarpi. Again, taxa that diverge strongly from the major axis of covariation are widely distributed across phylogeny, and birds which are relatively closely related may be outliers in different directions. We note that, while outliers have mixed phylogenetic affinities, a large overall proportion of outliers belong to the subclade Aequorlithornithes across all of our analyses.

## Discussion

The application of our tandem-method approach reveals an intersection between body mass evolution and the modular organisation of skeletal proportions that coheres with biomechanical expectations. Moreover, these findings lead us to generate new and original insights into adaptive radiation in birds that merit further pursuit. We show that body mass explains a component of modular integration of skeletal proportions across birds, with lower integration among the wing proportions in small birds and increasing integration between skeletal proportions within the wing and trunk. This structural re-organisation of evolutionary modularity in birds, reflecting greater consolidation of the wing among more massive taxa, accords with the biomechanical suggestions of previous authors that the constituent elements of load-bearing and locomotory structures should be more integrated in animals with larger body masses[28,43]. However, they contradict

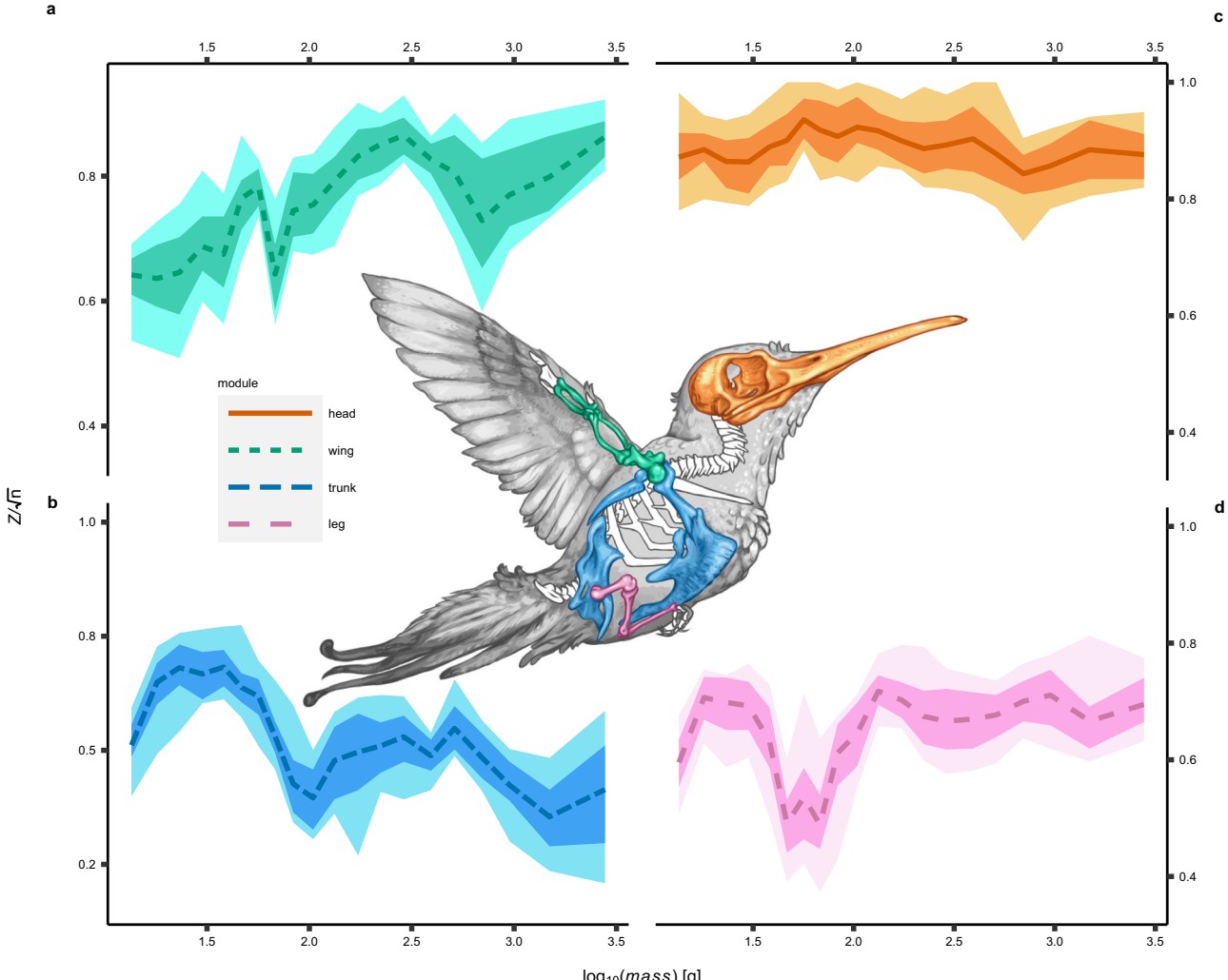

**Fig. 2 | Reorganisation of modular integration with varying body mass.**
**a** Change in within-wing integration ($Z/\sqrt{n}$) with body mass. **b** Change in within-trunk integration with body mass. **c** Change in within-head integration with body mass. **d** Change in within-leg integration with body mass. Effect sizes computed with phylogenetic 2-Block Partial Least Squares under permutation, and normalised by the root size of replicates. Sample size: $n = 228$ total species, grouped into 20 overlapping body mass bins. Integration statistics were then computed in $n = 30$ replicates within each bin, with each replicate subsampling $n = 30$ species. Dark and pale envelopes represent 1 and 2 $\sigma$ boot-strapped confidence intervals. Illustration A. Orkney 2023.

expectations that small-bodied passerine birds should exhibit a generalised pattern of locomotor modularity in which all modules are 'moderately developed', first posited by Gatesy and Dial[18]. The structure of modular aggregation and dis-aggregation we have uncovered here is disseminated across avian phylogeny (Fig. 4), which defies expectations that the emphasis of distinct combinations of modularity within a broader avian 'locomotor mosaic' should correspond strongly with avian subclades distinguished by unique combinations of ecology, body size and developmental mode (e.g. Dial, 2003 Figure 3[34]). We repeatedly found that the birds which differed the most from general axes of trait covariation, especially involving trunk elements, belonged to subclade Aequorlithornithes, echoing the findings of Navalón et al.[44]. Our observation that these birds tend to be outliers in multiple directions, even though they are relatively closely related, is also commensurate with Navalón et al.'s conclusion that ecomorphological convergence is commonplace within water birds.

We reflect that a range of additional anatomical characteristics beyond skeletal proportions will also necessarily influence the biomechanical performance of wings and their resistance to stress under high loading. Possible examples include postcranial pneumatisation, which is known to differ characteristically between diving and non-diving birds[45]; wing musculature and feathers, which exhibit allometric scaling while wing bone proportions do not[41]; and external wing-shape, which has a complex relationship to foraging behaviour across birds exploiting aquatic resources[46]. Histological properties such as bone density and microstructure are also likely to have important roles in ecomorphological adaptation, for example the proportion of laminar bone varies between the wing skeletons of birds practicing fast-flapping and dynamic-soaring flight-styles[47], and there is an observed correlation between sites of strain and remodelled bone deposition in the furculae of soaring birds[48]. We advocate the exploration of this additional suite of variables, and the ways in which they may intersect with skeletal proportions to mediate functional adaptation and mitigate mechanical stresses, as subjects of future research. In addition, we reiterate that 3D skeletal shapes in birds, which can have greater ecological relevance than proportions at clade-wide scales (e.g. refs. 33,49) do not exhibit a clear modular organisation.

We hypothesise that higher integration of skeletal proportions within the wing reflects the influence of tighter mechanical constraints at high body masses, reasoning that changes in the mechanical properties of one trait within a highly-constrained structure will require compensatory changes in other traits to maintain function. We suggest

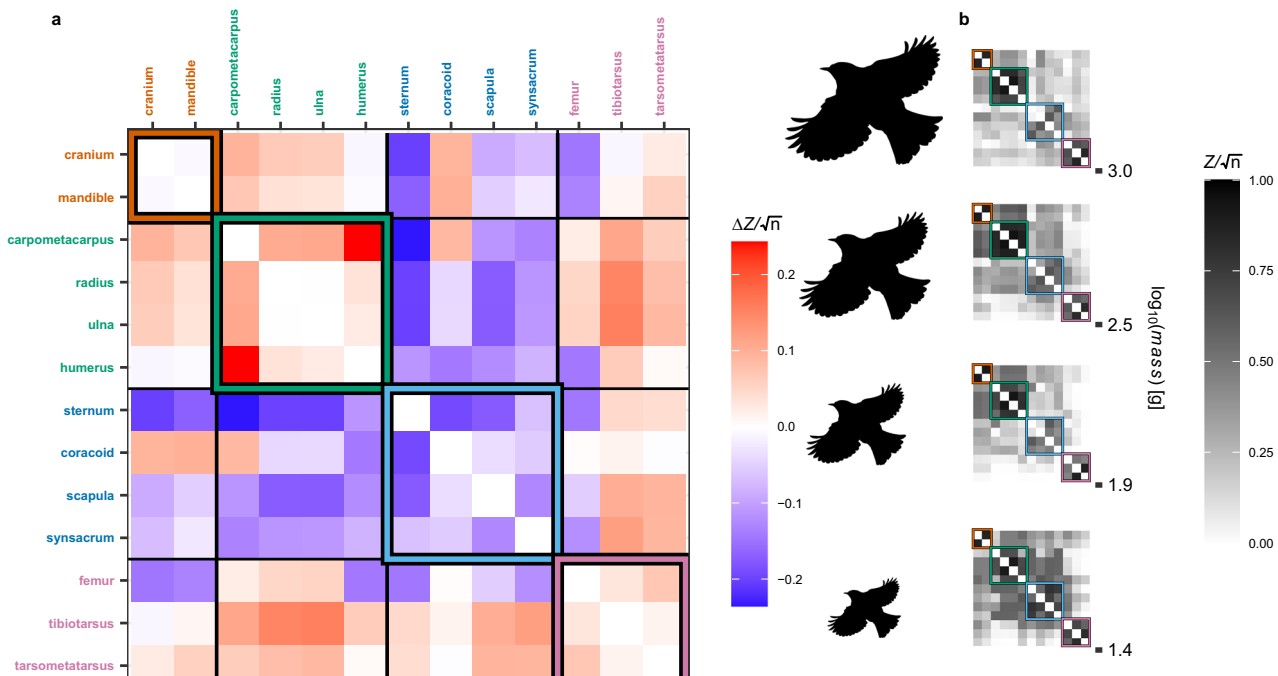

**Fig. 3 | Reorganisation of modular integration with varying body mass. a** Slopes of Ordinary Least Squares fits of integration between pairwise combinations of skeletal element sizes ($Z/\sqrt{n}$) depending on $\log_{10}(mass)$ (calculated across 20 bins). Skeletal modules are identified by colour; orange (head), green (wing), blue (trunk), pink (leg). Red/blue values indicate a tendency to become more/less integrated at higher body mass. **b** Visualisation of pan-skeletal pairwise integration values at body mass bins ranging from 13–36 g, 54–124 g, 163–456 g and 456–1724 g. Sample size: $n = 228$ total species, grouped into 20 overlapping body mass bins. Integration statistics were then computed in $n = 30$ replicates within each bin, with each replicate subsampling $n = 30$ species. Ordinary Least Squares models were then fit to the integration statistics imputed across the 20 bins. The silhouette of *Turdus pilaris* was procured from phylopic.org, (Sharon Wegner-Larsen), under a CC0 1.0 Public Domain license.

that evolutionary novelty within small-bodied clades, such as Apodiformes (the subclade containing swifts, hummingbirds and treeswifts), may have been permitted by decreased within-wing integration as a result of their small body size, allowing the independent adaptation of individual wing elements to accommodate the exploration of new flight-styles and distinctive wing skeletons[50]. Indeed, previous work has shown that the relative size of the carpometacarpus– which is relatively enlarged in Apodiformes– is a strong predictor of flight-style variety across birds[33], and we observe that the divergent evolution of the carpometacarpus is a principal driver of the patterns we have presented here. We can conceptualise these findings as a 'trade-off' between the independent adaptation of constituent elements of the wing– more typical of small birds– and their harmonious integrated function in more massive birds operating under more restrictive mechanical constraints. Conversely, constant levels of within-head and within-leg integration with increasing body mass may reflect the fact that these structures do not take preeminent load-bearing roles in forelimb-propelled birds. The leg and head hence experience no trade-off requiring the increased consolidation of elements proportions to resist greater mechanical stress at the expense of evolvability in more massive birds. We may therefore expect ecomorphological evolvability of these structures to be high across forelimb-propelled birds. Indeed, birds exhibit very high evolutionary rates and a broad disparity of hindlimb proportions, when contrasted with their heavier, terrestrial non-avian dinosaurian relatives[30]. We predict that large flightless birds should exhibit stronger evolutionary integration among leg bone sizes, and hypothesise that they will also explore a smaller repertoire of leg proportions.

We suggest that songbirds play a substantial role establishing strong evolutionary integration between the skeletal proportions of the wing and trunk in small-bodied birds. Indeed, passerines constitute the majority of small birds, both in our dataset and across extant avian diversity (see Fig. 1). We suggest that the relative sizes of bones within the trunk and wing may be more integrated among small birds, and especially passerines, because they become unified as a single locomotory module in intermittent and flap-bounding flight-styles, which are restricted to small birds. Flap-bounding flight, in which a bird temporarily travels under ballistic motion between brief bursts of intense flapping[50], requires the wing to lie flush against the trunk to form a fusiform lift-generating surface. This could mean that, in flap-bounding birds, the wing and trunk dimensions are required to match and that their capacity to evolve independently is diminished compared to other birds.

We suggest this requires the coupling of phenotypic variation in the trunk and wing, and note that the upper limit for flap-bounding flight is estimated at ≈300 g,[50,51]. Birds larger than this mass tend to exhibit lower levels of within-trunk integration (Fig. 2b) and birds with the most divergent carpometacarpus-sternum size ratios also tend to possess masses above 300 g (Fig. 4e). It may therefore be possible that fundamental shifts in the flight-styles that are possible at different body masses underlie shifts in modular integration across birds, which suggests that– beyond determining evolutionary lability– evolutionary modularity may itself respond to adaptive demands in birds.

Outlying taxa within the trunk-integration space (Fig. 4d) include a selection of semiaquatic birds. Previous researchers have reported that aquatic birds may specifically emphasise the relative sizes of different components within the trunk depending on their specific locomotory mode (e.g. foot or wing-propelled diving), and furthermore surface-swimming birds exhibit a wide variety of skeletal proportions within the trunk[52]. We suggest that this disparity may explain the preponderance of birds in clade Aequorlithornithes that are outliers across our trunk integration analyses (Fig. 4d, e). Additionally, buoyant support of the trunk in water might alleviate selection pressure acting against unusually massive sterna, expanding the viable trait

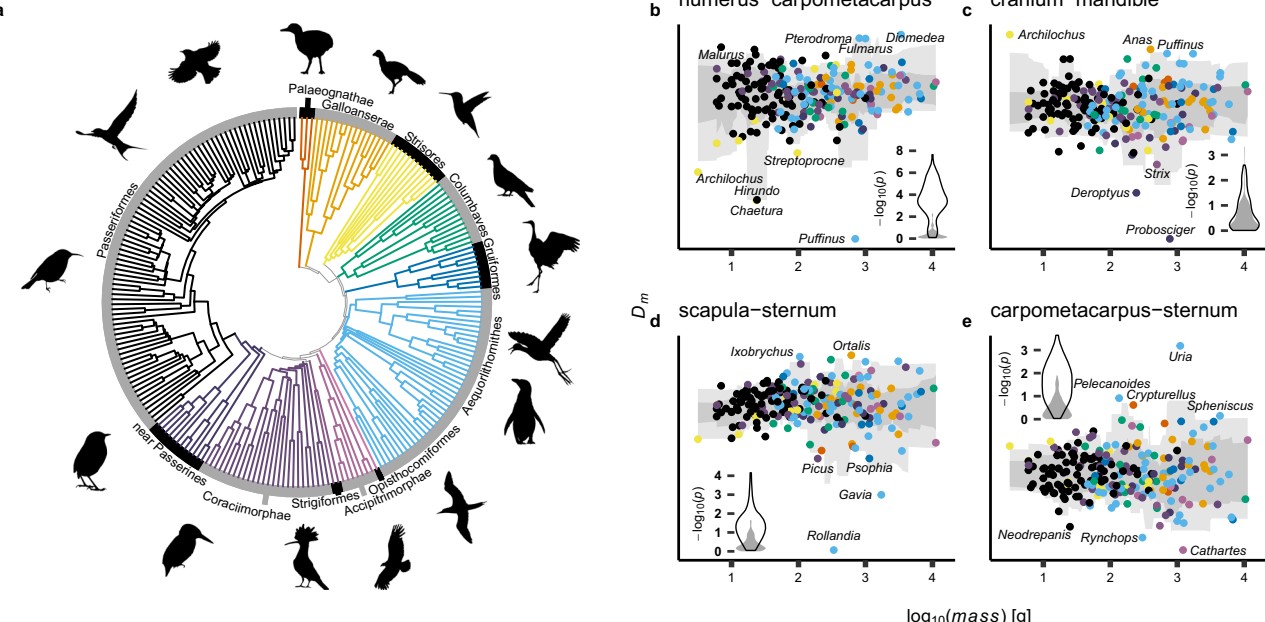

**Fig. 4 | Phylogenetic structure of modular reorganisation. a** Phylogeny with species binned into $k = 12$ coloured groups to aid visualisation of species relatedness in subsequent plots. Plots of the second major axis of integration ($D_m$; inversely related to integration strength) and its dependency upon body mass between chosen pairs of skeletal elements: carpometacarpus-humerus (**b**), cranium-mandible (**c**), scapula-sternum (**d**) and carpometacarpus-sternum (**e**). Light and dark grey ribbons represent 1 and 2 $\sigma$ probability intervals over an $n = 30$ species running quantile, respectively. Taxa are coloured to match the phylogeny in sub-plot (**a**). Inset flame-plots indicate $p$-values associated with one-tailed Breusch–Pagan tests for heteroskedasticity in re-sampled data (black contour), compared to a null-distribution of permuted data (grey). Sample size: $n = 228$ species. Re-sampling was conducted 100 times, by dividing the range of body masses into 10 equal bins, and subsampling up to 10 species within each bin to produce a subsample with a Gaussian distribution of body mass values. Individual subsample sizes were allowed to vary, so that a variety of plausible Gaussian

distributions could be explored and to ensure that variations in taxonomic sampling completeness across the avian tree do not confound analyses. No correction is performed for multiple comparisons, which necessitates the production of the null distribution for comparison. Silhouettes were sourced from phylopic.org. *Acanthorhynchus tenuirostris* (Michael Scroggie), *Alcedo atthis* (Margot Michaud), *Archilochus colubris* (Andy Wilson), *Grus canadensis* (Sharon Wegner-Larsen; reflected horizontally), *Hirundo rustica* (Matt Wilkins), *Jacana spinosa* (Burton Robert; reflected horizontally), *Spheniscus humboldti* (Juan Carlos Jerí), *Turdus pilaris* (Sharon Wegner-Larsen), *Upupa epops* (Ferran Sayol) and *Vultur gryphus* (Ferran Sayol) were procured under a CC0 1.0 Public Domain license. *Columba palumbus* (Anthony Caravaggi), *Sula sula* (Larry Loos (photography) and Michael Keesey (vectorisation)) and *Xenicus gilviventris* (Wynston Cooper (photo) and Albertonykus (silhouette)), were procured under a CC BY-NC-SA 3.0 DEED license. *Crax alector* (J. N. Wiegers; reflected horizontally) and *Crypturellus variegatus* (J. N. Wiegers) were procured under a CC BY 4.0 DEED license.

space accessible to water birds. We also reflect that previous work has demonstrated that 3D sternal shape adapts to a wide variety of ecological demands across birds[33,53,54], which may not be captured in a study that considers skeletal element sizes alone.

We found systemic increases in the variability of skeletal proportions among more massive birds (Fig. 1b). This finding is compatible with previous observations by Hallgrímsson and Maiorana[55], which they attributed to the tendency for the contribution of skeletal tissue to organism mass to vary more than other tissue types, such as nervous tissue or viscera. Skeletal tissue tends to contribute a larger proportion of overall body composition in heavier animals, and this may contribute to greater variability at an interspecific scale in the allometry-corrected proportions of large birds' skeletons. Additionally, we hypothesise that trunk skeletal elements in particular may conform more closely to the proportions of internal organs in small birds that may be approaching the physical limits of miniaturisation, which we suggest may temper the variability of trunk skeletal element sizes in comparison to larger bodied taxa. If correct, this hypothesis predicts stronger integration between soft tissue mass and trunk skeletal mass in smaller birds, both at an intra and interspecific scale. The variability of trunk skeletal elements was the lowest of any skeletal module, while the leg exhibited the highest values.

The observation that the relative sizes of more distal skeletal elements within the head and leg tend to be more variable (Fig. 1b; Head, Leg) is consistent with predictions under a proximo-distal gradient of embryogenesis[56] and previous observations of greater

evolutionary variance in more rostral components of the avian cranium[8]. The relatively high phenotypic disparity of peripheral skeletal proportions, compared to the proximal skeleton, could also reflect the fact that these are the elements interacting most directly with the substrate of the external environment- accommodating the manipulation of food (mandible, rostrum and feet), arboreal, terrestrial, and aquatic locomotion (feet). These peripheral elements may therefore simply be more amenable to ecomorphological selection, as suggested by Orkney et al.[33]. Authors such as Navalón et al. have also suggested that non-passerine land-birds exploit the adaptation of peripheral skeletal elements to mediate ecological adaptation[44].

The wing is a notable outlier in our analysis, with no clear proximo-distal gradient in the variability of the relative size of different skeletal elements. We suggest that strong evidence of the pervasive biomechanical constraint of this structure in fore-limb propelled birds (see Figure 4 Navalón et al.[44]) requires the consorted evolution of wing proportions as a single functional complex, tempering the proximo-distal gradient. This may explain why, although carpometacarpus size is strongly relevant to flight-style across birds, its 3D shape– which might not be evolving under the same constraints– exhibits an even clearer relationship to external wing geometry and flight-style variety[33,49]. Our result here is distinct from observations in mammals[57], in which forelimb evolution exhibits a strong proximo-distal gradient of morphological disparity. This departure in birds from a pattern reflecting the underlying embryogenetic sequence lends further credence to the hypothesis that integration within the avian

wing may be under strong mechanical constraint that causes a unique variance structure and evolutionary coalescence around a small range of wing proportions. Indeed, avian wing proportions have been shown to scale isometrically with body mass, which stands in strong contrast to their hindlimb evolution and implies the conservation of an optimum kinematic ratio across most birds[42]. Additional support is lent to this hypothesis by our finding that the relative size of the scapula, which has a direct role in the avian flight apparatus, is not characterised by greater levels of variance compared with other trunk elements, despite occupying a more peripheral position within the trunk.

We reflect on the possible generalisations of the statistical approach we have presented herein. We observe that bats are distributed across three orders of magnitude of body mass and that they face similar biomechanical and functional constraints to birds. Previous research has suggested that the bat limb skeleton exhibits a similar profile of dissociated fore and hindlimb evolution to birds[58] and we might ask whether there are further examples of convergence mediating the adaptation to body mass variety in these clades and- if so- whether this might indicate a common set of macroevolutionary rules that govern vertebrate flight. The variety of potential questions that could be engaged with heuristic analysis of $D_m$ could be extended beyond animals, to investigate whether evolutionary covariances in plants are re-shaped by biomechanical demands in diverse radiations that have a wide distribution of masses (e.g. Fabaceae, Rosaceae). Of course, beyond body mass, the researcher might select any variable by which to explore deviation from major axes of trait covariance, such as bite-force, maximum running speed, or diving depth. We caution that our approach assumes that skewness in the distribution of the independent variable (in this case body mass) can be neglected, which required a bespoke re-sampling scheme and the computation of null distributions of statistics (see methods). Our approach could also be generalised to categorical data to explore whether exclusive ecological variables are significantly associated with changes in evolutionary covariance. For example, a Brown–Forsythe test for unequal $D_m$ could be applied to determine whether saltatorial or fossorial rodents' hindlimbs are more or less integrated than those belonging to terrestrial forms.

We conclude that our analysis of evolutionary covariances between the relative sizes of avian bones shows that the avian wing is a uniquely integrated structure within the avian body plan. We hypothesised that the strong consorted evolution of bones within the avian wing should be amplified under high body mass, which places tighter mechanical demands on this structure, and alleviated in birds with lower body mass. Our findings support this hypothesis, demonstrating that bones within the wing evolve more freely of each other in smaller birds. We also found that the evolution of wing bones becomes increasingly allied with the trunk in smaller birds. The evolutionary implications of this modular reorganisation are diverse. We suggest that weakened integration within the wing has facilitated the discovery of new wing arrangements and mechanical optima in apodiform birds, which have a disparate array of wing proportions (e.g. swifts, hummingbirds). By contrast, the evolution of skeletal proportions is more conservative in songbirds. We suggest that the coupling of wing and trunk evolution in this group facilitates their bounding flight-styles, which are restricted to small-bodied taxa, and that the ecological opportunities offered by passerine-style locomotion in cluttered environments may represent a key-innovation integral to their success in Cenozoic closed-canopy settings (e.g. ref. 59).

## Methods
### Data
Original 3D shape data were obtained from constellations of homologous landmarks, which consist of an ecologically disparate and phylogenetically comprehensive sample of 228 extant bird species[44].

The centroid sizes of major skeletal elements (cranium, mandible, scapula, coracoid, sternum, synsacrum, humerus, ulna, radius, carpometacarpus, femur, tibiotarsus, tarsometatarsus) were obtained from a generalised Procrustes alignment computed in the R *geomorph* package[60–63] version 4.0.5 with the gpagen() function. Centroid sizes were log$_{10}$-transformed and employed to represent the sizes of skeletal elements. Data were placed in a phylogenetic context using a fossil-calibrated phylogeny compiled by Navalón et al.[44], which was produced by combining the phylogenies of Prum et al.[64] and Oliveros et al.[65]. We used phylogenetic structure to account for statistical non-independence[66]. Body masses were obtained from the metadata of Navalón et al.[44]. These values were originally qualified with reference to the Cornell Handbook of the Birds of The World[35], and when available the average male and female body mass was used to estimate the mass of each species, given that the sex of individual birds was often unknown.

### Analysis
We develop a simple approach that can be widely generalised, applying tandem methods to assess whether interspecific integration of skeletal element sizes within major modules (head, wing, trunk, leg)[18,33,58,67] and the overall scheme of modular organisation of the avian skeletal proportions (the pattern of pairwise integration across the body plan[68,69]) vary with body mass. We explored integration of structural regions within taxonomic subsets of varying body mass (method 1), and investigated dispersion of individual taxa from major axes of trait covariance ('$D_m$'; method 2).

**Initial appraisal of variance structure.** We initially assessed the degree of phylogenetic signal in body mass variation across our sample, to establish that any resultant findings are not driven primarily by phylogenetic pseudo-replication. We computed Blomberg's $K$ and Pagel's $\lambda$ statistics for log$_{10}$(mass) across the 228 bird taxa. Blomberg's $K$ represents the ratio of among-species variance, compared to the variance expected under Brownian evolution; values much greater than 1 therefore indicate strong phylogenetic autocorrelation. Pagel's $\lambda$ represents phylogenetic autocorrelation between species, with values close to zero representing total independence, while values near 1 are compatible with a Brownian model of evolution. Statistics were computed with the phylosig() function from the *phytools* package[70] version 1.5-1.

Thereafter, we computed allometrically-corrected centroid sizes for skeletal elements with the gls() function of the *nlme* package[71] version 3.1-162, representing phylogenetic autocorrelation with a corPagel autocorrelation structure initiated with $\lambda$ values imputed by the phylosig() function. We computed the difference between the first and fourth quantiles of variance within each skeletal element over a running window of 30 taxa, using the runquantile() function of the *caTools* package[72] version 1.18.2, setting the end rule to 'quantile'. This allowed us to inspect how interspecific variance in skeletal element proportions varies between small and large birds.

We imputed an ancestral state reconstruction of log$_{10}$(mass) across our sample with the fastAnc() function of the *phytools* package, which we used to identify avian sub-clades that are typified by unusually small and potentially derived body sizes.

**Method 1: Phylogenetic partial least squares of Skeletal Element Sizes.** We computed indices of evolutionary integration between all pairwise combinations of skeletal element sizes ($n = 78$) in cohorts of bird species with different mean body masses, in order to explore whether the evolutionary organisation of the avian skeleton changes with body mass. We constructed a scheme of 20 body mass bins, which partially overlapped, to achieve this. We ordered all 228 species in the dataset by mean body mass. We stated that the first bin is constituted by the smallest 40 taxa. Thereafter, we moved this

window of 40 taxa by 9 or 10 birds, so that the next bin was constituted by all bird species whose body masses rank between the 10th to the 50th smallest taxa. This process was repeated until all 20 bins were defined.

We then randomly selected 30 taxa within each bin, and computed allometric models of skeletal element sizes, depending on $\log_{10}(mass)$, with the gls() function of the $nlme$[71] package, employing the corPagel autocorrelation structure to represent the effects of phylogenetic non-independence. We initialised these models with $\lambda$ values estimated by the phylosig() function of $phytools$. We extracted model residuals for the subsequent calculation of integration statistics. Our decision to divide the dataset of 228 birds into many bins means that allometric models are fitted over relatively small ranges of body masses, which accommodates the possibility that allometric scaling can itself vary across clades[73].

We employed a phylogenetic two-block partial least squares approach[26,74], implemented with the phylo.integration() function of $geomorph$ to compute Z-scores (effect-sizes) of integration between the 78 pairs of skeletal elements. We repeated this process 30 times, so that different permutations of species were sampled. After this process was complete, we normalised all mean Z-scores by $\sqrt{30}$, because Z-scores calculated in $geomorph$ are a function of the root of the sample size. We then set all negative $Z/\sqrt{n}$ values to zero, before averaging all $Z/\sqrt{n}$ values belonging to pairwise combinations within pre-defined modules of the avian body plan (head, wing, trunk, leg). This process was repeated for each body mass bin. We then applied a bootstrapped procedure with 1000 re-samplings to produce a probability distribution of the mean value of $Z/\sqrt{n}$, allowing us to construct a confidence interval around our results. We thus produced a series of possible 'within-module' integration statistics for 20 different bins of avian body mass, allowing us to assess how evolutionary integration within the head, wing, trunk and leg changes between the smallest and largest birds.

Additionally, we fitted linear ordinary least squares models of mean $Z/\sqrt{n}$ values across the 20 mass bins against the vector of 20 $\log_{10}(mass)$ values. We performed this procedure for each pairwise combination of skeletal elements. We fitted these models with the lm() function of base R $stats$ package version 4.2.3[75]. The slopes of these models provide a qualitative impression of the leading pairwise combinations of skeletal element sizes that cause changes in modular integration between small and large birds.

**Method 2: Heteroskedasticity of taxa and distances from axes of trait covariance.** We then explicitly examine the individual contributions of taxa to gross shifts in modular organisation with body mass by exploring how body mass structures the dispersion of individual taxa from major axes of trait covariance. This method makes different assumptions about the role of phylogeny and permits significance testing. A common allometric model over all considered taxa is assumed, and statistical analyses are undertaken in a phylogenetic context that accommodates all taxa simultaneously.

We computed allometrically-corrected centroid sizes for skeletal elements with the gls() function of the $nlme$ package, representing phylogenetic autocorrelation with a corPagel correlation structure initiated with $\lambda$ values imputed by the phylosig() function of $phytools$, and extracted model residuals for subsequent analysis. We then computed phylogenetic two-block partial least squares analyses for humerus size-carpometacarpus size, sternum size-scapula size, and sternum size-carpometacarpus size with the phylo.integration() function of $geomorph$, and found the major axes of covariance ($m$) that related each pair of skeletal elements with the prcomp() function of the $stats$ package of base R, and extracted the Euclidean distances of each species to the major axis of covariance, $D_m$ (equivalent to the second principal component in a bivariate system). A reduction/increase in the dispersion of $D_m$ with greater body mass implies increasing/decreasing integration in more massive species. We plotted

$D_m$ as a function of $\log_{10}(mass)$ to visualise how integration changes with body size, and identify individual aberrant taxa or avian sub-clades. We chose to investigate these pairs (humerus-carpometacarpus, sternum-scapula, sternum-carpometacarpus) because results from method 1 indicate they show strong changes in integration with body mass (Fig. 3) and we can thus make clear predictions about the structure of their heteroskedasticity (declining heteroskedasticity with body mass between the carpometacarpus and humerus, and increasing heteroskedasticity between the sternum and scapula and the sternum and carpometacarpus).

Thereafter, we sought to assess whether patterns we could identify visually should be considered significant, which necessitated a re-sampling scheme that could ensure that outliers and a right-skewed distribution of $\log_{10}(mass)$ did not have an out-sized influence on results. We therefore defined 10 evenly-spaced breaks in $\log_{10}(mass)$, and randomly sampled up to 10 taxa between each break, to produce sub-samples which converged on a Gaussian distribution of $\log_{10}(mass)$. We then repeated our approach to calculate $D_m$, fitted a linear ordinary least squares regression of $D_m$ depending on $\log_{10}(mass)$ with the lm() function of $stats$ and performed a Breusch–Pagan test with the ncvTest() function of the R $car$ package[76] version 3.1.1 to determine whether heteroskedasticity of the residuals depends upon $\log_{10}(mass)$. We repeated this process 100 times to generate a distribution of $p$-values, and also generated a distribution in a permuted dataset in which all skeletal element sizes and body mass values have been randomly-reordered. We then super-imposed the negative logarithm of the real sub-sample and null distribution $p$-values to produce 'flame-plots' that indicate significance within a robust re-sampled framework. Highly significant relationships will result in flame-plots that exhibit a real sub-sample distribution that extends to larger values than the null distribution, whereas non-significant relationships will result in overlapping real and null distributions.

We checked that a tendency for increasing skeletal variance among more massive taxa (Fig. 1b) did not confound our analyses by performing a test for heteroskedasticity between cranium and mandible size (Figs. 2d and 4c). We expected these structures to be strongly integrated in both small and large birds, and therefore predicted that there should be no significant change in integration between these features across body mass. Our results confirm that integration between the cranium and mandible remains constant, and that systemic changes in skeletal variance with body mass do not confound our analyses (Fig. 4c).

The $abind$ package version 1.4-5[77], $ape$ package version 5.0[78], $dplyr$ package version 1.1.1[79], $reshape2$ package version 1.4.4[80] and $zoo$ package version 1.8-12[81] were used to curate and format data. The $cowplot$ package version 1.1.1[82], $dendextend$ package version 1.17.1[83], $geomtextpath$ package version 0.1.1[84], $ggdendro$ package version 0.1.23[85], $ggnewscale$ package version 0.4.9[86], $ggplot2$ package version 3.4.1[87], $ggpubr$ package version 0.6.0[88], $ggrepel$ package version 0.9.3[89], $grid$ package of base R, $jpeg$ package version 0.1-10[90], $magick$ package version 2.8.2[91] and $png$ package version 0.1-8[92] were used to produce visualisations.

### Reporting summary

Further information on research design is available in the Nature Portfolio Reporting Summary linked to this article.

## Data availability

Original skeletal landmark constellations were sourced from Navalón et al.[44]. The original phylogenies used to represent evolutionary non-independence was sourced from Prum et al.[64] and Oliveros et al.[65]. Figures presented in the paper herein can be generated from the codes available in our GitHub repository: https://github.com/aorkney/TinyBirds Zenodo deposition: https://doi.org/10.5281/zenodo.10879690.

## Code availability

DOI numbers indexing original datasets, and custom code required to duplicate the analyses and reproduce the figures herein are found at our GitHub repository: https://github.com/aorkney/TinyBirds Zenodo deposition: https://doi.org/10.5281/zenodo.10879690.

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

## Acknowledgements

We acknowledge Doctor Alex Bjarnason and Doctor Roger Benson, who constructed much of the original dataset of skeletal landmark constellations and who have provided continued support to facilitate its investigation by other scientists. We acknowledge Doctor Guillermo Navalón for his efforts to expand this dataset and his assistance parsing these data before final analysis. We acknowledge the following images from https://www.phylopic.org/: *Acanthorhynchus tenuirostris* by Michael Scroggie (CC0 1.0 Universal Public Domain Dedication) *Alcedo atthis* by Margot Michaud (CC0 1.0 Universal Public Domain Dedication) *Archilochus colubris* by Andy Wilson (CC0 1.0 Universal Public Domain Dedication) *Columba palumbus* by Anthony Caravaggi (reflected around x-axis) (https://creativecommons.org/licenses/by-nc-sa/3.0/) *Crax alector* by J. N. Wiegers (reflected around x-axis) (https://creativecommons.org/licenses/by/4.0/) *Crypturellus variegatus* by J. N. Wiegers (https://creativecommons.org/licenses/by/4.0/) *Grus canadensis* by Sharon Wegner-Larsen (reflected around x-axis; CC0 1.0 Universal Public Domain Dedication) *Hirundo rustica* by Matt Wilkins (CC0 1.0 Universal Public Domain Dedication) *Jacana spinosa* by Burton Robert (reflected around x-axis; CC0 1.0 Universal Public Domain Dedication) *Spheniscus humboldti* by Juan Carlos Jerí (CC0 1.0 Universal Public Domain Dedication) *Sula sula* by T. Michael Keesey (vectorization) and Larry Loos (photography) (https://creativecommons.org/licenses/by/3.0/) *Turdus pilaris* by Sharon Wegner-Larsen (CC0 1.0 Universal Public Domain Dedication) *Upupa epops* by Ferran Sayol (CC0 1.0 Universal Public Domain Dedication) *Vultur gryphus* by Ferran Sayol (CC0 1.0 Universal Public Domain Dedication) *Xenicus gilviventris* by Wynston Cooper (photo) and 'Albertonykus' (silhouette) (https://creativecommons.org/licenses/by-sa/3.0/).

## Author contributions

A.O. downloaded and prepared skeletal and requisite metadata, generated figures and illustrations. A.O. and B.P.H. jointly designed, undertook analyses and wrote the manuscript.

## Competing interests

The authors declares no competing interests.
