## [Peer Review File · Nature Communications]

Small body size is associated with increased evolutionary lability of wing skeleton proportions in birdsREVIEWER COMMENTS

Reviewer #1 (Remarks to the Author):

This manuscript evaluates the link between body mass and the modularity (i.e., the structure of trait covariation (i.e., integration)) of size proportions across the skeleton in a rich sample of birds (although see comments below for improvement). The major finding of the study is an intriguing relationship between small body sizes and weaker size proportions integration in the wing skeleton which the authors explain as, mostly, reflecting the release of biomechanical constraints of flight associated with larger birds, allowing small birds to explore other optimal solutions to the construction of the wing (e.g., hyperaerial apodiiforms and hirundinids). The manuscript also proposes a novel methodological pipeline that opens interesting avenues for future research. This manuscript excels in some ways such as the analytical component or the introduction to the topic of modularity and its generative factors at several evolutionary levels. However, I think there are several aspects that can be improved to enhance the accessibility of the findings overall. Furthermore, I have some additional concerns.

First, I think it is important to clarify throughout the manuscript that when the authors are talking about integration and modularity this refers to relative sizes of skeletal elements. This must be clear from the title and the abstract. Personally, it took me a while to understand that the authors were not talking about shape/form integration.

Second, is there any reason why the reduced version of 'TEMPO BIRDS' is used instead of the largest and most updated version of this dataset (i.e., Navalón et al., 2022)? The expanded version includes ~100 species many of which are passerines – a group including among the smallest birds and widely discussed in the present manuscript – but also some of the smallest representatives of other lineages (e.g., *Micropsitta*, Psittaciformes). Unless there is a major reason for not including this readily available dataset, I encourage the authors to make full use of the available data. If the reason – as I suspect – is the unavailability of flight style traits for the expanded dataset, then I reckon that: 1) this data mostly exists in large global datasets like AVONET; 2) this latter part of the manuscript is such a minor aspect of the current study that it does not overweight the benefits of including a much richer dataset of smaller birds, and 3) links between body mass and species richness have been widely explored and probably (as admitted by the authors) are explained by many other factors related with generation times, metabolism, etc. Therefore, I recommend that even if the authors do not include the richer dataset that is readily available (which I encourage), excluding connections between flight style, richness and body mass will streamline an already (conceptually and analytically) complex paper. To be clear I think they could still discuss aspects of flight style biomechanics in the discussion which in reality they are not derived from their analyses.

More specific comments:

Introduction:

I can be wrong but I think the actual bird species count is closer to 11K than to 10K.

Methods:

You need to state all the R functions and packages that you used.

Discussion:

A number of findings here resonate with the findings in Navalón et al., 2022. Including a clear environmental structure in patterns of morphological disparity across the skeleton – including but not restricted to skeletal proportions (e.g., much larger disparity of skeletal proportions in waterbirds with respect to landbirds).

'268 the highest values. The observation that more distal skeletal elements within the head and leg

269 tended to be more variable' This result can be also interpreted as the effects of adaptation in the elements of the avian body that are in closest contact with the environment and therefore are more heavily shaped by selective pressures related to ecological traits as proven by an previous paper lead by the same first author (Orkney et al., 2021)

I think the discussion could benefit from a summary statement that links back to the main evolutionary component of the study (i.e., links between body mass and the integration in skeletal size proportions)

Figures:

Figure 2c could be a different figure together with a version of Fig.2a with the inset heatmaps. Resulting Figure 2 would retain a,b,d,e and the schematic skeleton excluding the insets from 2a.

Figure 3. Some labels for clades and silhouettes to guide the reader's attention. Same for Figure 1. This will make the paper more appealing to a wider readership like the beautiful drawing of Figure 2.

Reviewer #2 (Remarks to the Author):

In this manuscript, Orkney and Hedrick use a dataset comprising morphometric data for a variety of skeletal elements collected from 149 volant bird species to investigate the evolutionary allometry of the functional modularity of the avian skeleton. The authors found that, in general, variability in skeletal element size increased with increasing body mass (especially more distal elements), as did strength of integration among the module comprising the wing elements; by contrast, strength of integration among the module comprising the trunk elements as well as between the wing and trunk modules decreased with body mass. Those patterns were supported by pairwise comparison of selected elements, revealing that increasing body mass was linked to tighter integration between humerus and carpometacarpus size, but weaker integration between the scapula and sternum and the carpometacarpus and sternum.

The patterns the authors seek to elucidate here are fundamental ones in bird evolution, and so this paper is likely to draw the interest of a wide range of functional, organismal and evolutionary biologists. The data collected and analyses used seem sufficient to investigate the allometric evolution of the sampled skeletal elements across birds, and the results a quite fascinating. I applaud the authors for making all the data and scripts that went into their analyses available – if only all authors were as forthcoming!

I have several concerns about the authors' conclusions – in particular, about the assumptions made by the authors about the biological significance of their chosen data. There are also some parts of the paper I found quite confusing and unclear – specifically, the authors' description of "method 1" and the discussion of "Pareto fronts."

My first major concern regards the biological significance of the authors' analyses. What the authors are specifically investigating is the (co)evolution of the allometry of the sizes of the skeletal elements sampled here. From there, the authors implicitly assume that there is a meaningful link between, on the one hand, evolutionary shifts in those allometries, and, on the other hand, functional shifts in some aspects of the modules comprising those elements. However, the authors do not explain why we might expect such a relationship, let alone what that relationship might be. To be clear, I am *not* suggesting that such a relationship does not exist – indeed, I find it quite intuitive that it would – but something approaching a clear, explicit understanding of that relationship must be provided by the authors to support any conclusions drawn by the analyses presented in this paper. I strongly urge the authors to establish this expected link in the introduction and explore what their results might tell us about that link in the discussion.

Relatedly, when discussing the patterns they reconstruct, the authors must acknowledge the myriad other traits that might influence those patterns besides total size. Indeed, many other aspects of a given skeletal element are likely to influence the function of that element, arguably with greater strength – for example, degree of pneumatization, the relative development of sites for muscle attachment, or bone thickness and microstructure. Furthermore, many aspects of other, related tissues are likely to major influence the evolution of skeletal modules – for example, associated musculature, and functionally related organs. This is especially relevant to the wing, to which the authors understandably devote much of their discussion. Indeed, one might easily imagine shifts in overall airfoil shape, feather distribution, and feather morphology profoundly influencing the function of the wing module without any appreciable shift in the sizes of the underlying skeletal elements. The authors do mention such potential influences in select cases, such as the sternum (but see my comment below), but these factors are largely ignored. I understand that testing for concomitant shifts in those other traits was beyond the scope of this

study, but their exclusion must still temper the conclusions drawn by the authors from their analyses. I find it possible – perhaps likely – that quite different patterns might be recovered if other aspects of this system were sampled, instead. For example, I suspect that the shapes/morphologies of the trunk elements (e.g., the functional surfaces of the scapula and coracoid, the sizes of the keel and notches of the sternum) are far more directly linked to function than size, and thus might show a significantly different pattern of integration between the wing and trunk modules. The potential – indeed, likely – confounding intersection of these many traits should be discussed by the authors throughout their paper.

In many cases, the conclusions drawn by the authors are dependent on assumed evolutionary patterns that are not supported in this paper. The most crucial example is the pattern invoked in the title of the paper: miniaturization within birds. In many cases (most notably among passerines and hummingbirds), the authors explicitly assume that a lineage has been marked by miniaturization but provide no support for that assumption. In the introduction, the authors support the statement that birds have been “critically defined by miniaturization” (lines 63-64) by citing Benson et al. (2018), but that paper investigated trends in body mass leading to the stem lineage of extant birds, not among extant birds proper. If the authors were referring to papers cited within Benson et al. (2018), they should cite those papers, instead. If the authors mean to invoke their own analysis of body mass evolution, they should provide relevant discussion of those patterns. Nevertheless, the authors should provide support for any pattern of miniaturization they infer to justify the invocation of such a pattern in the title.

In other cases, the authors do support invoked patterns with citations, but they are not clear about what those patterns actually are or how they might be connected. For example, on lines 260-263, the authors compare their recovered pattern of increasing variability in skeletal proportions with increasing body size to a previously reported pattern of “high inherent variability of skeletal tissue.” As written, without going to the paper cited, I do not have a clue what “high inherent variability” means, and so I have no idea what the authors are trying to say here. On lines 263-266, the authors propose a stronger functional link between the sizes of trunk elements and internal organs among smaller birds than larger birds without explaining why we might expect the organs to exert pressure on the sizes of those elements (especially the elements besides the sternum), and they do not provide a citation for sized-dependent variability in size of internal organs. On lines 242-244, the authors propose a functional link between trunk and wing integration and flight style specifically among passerines, but they do not explain what that link might be.

In lines 237-238, the authors suggest that the trunk does not “have an obvious locomotory function.” This is not true at all! The scapula, coracoid, and sternum are directly involved in the flight stroke!

I did not find any conclusions drawn from the authors’ analyses of species richness to be compelling, and I think that aspect can be cut from the manuscript without any cost to the utility of the study – which I urge the authors to do. A sample of only 149 species out of the more than 11,000 known species is sufficient to investigate the evolution of traits like skeletal element size because it can reasonably be assumed that one or few species are a reliable enough proxy for the condition characterizing the entire lineage to which those species belong. However, I am highly skeptical that the same can be said when species number is the very trait being investigated. Minimally, the authors should provide a metric of how reliably their sample serves as a proxy for total species richness and discuss potentially consequences of their low sample size, but again, I think all discussion of this can and should be removed.

Similarly, I am not compelled by discussion of recovered patterns of ecological evolution beyond potential links to the wing. The authors’ chosen proxy for ecology is flight style, which we might expect to be mostly linked to the wing and potentially to the trunk, but only weakly linked to the skull and leg.

I found the discussion of “Pareto fronts” (paragraph 3 of the discussion) to be largely impenetrable. What I *think* they authors were trying to communicate is: “If the values of traits of interest are strongly linked among elements, then forces constraining the value of one such trait

will also constrain the values of traits to which it is linked." If my read there is incorrect, I urge the authors to "reverse engineer" how I might have come to such a misunderstanding. If my read is correct, it seems to me that this paragraph is largely redundant with the preceding paragraph, and so I recommend the authors fold whatever point they sought to make in this paragraph into that one. Minimally, the authors should reevaluate how they are approaching this point. Either way, I strongly urge the authors to remove direct discussion of "Pareto fronts" completely and to limit indirect discussion to a citation – it seems to me that, in order to clearly describe what they are and how they pertain, the authors will need to devote far too many column inches to the concept, distracting the reader from what we should be focused on.

Similarly, I found part of the description of "method 1" unclear. The authors did a fine job of outlining their treatment of body size bins and their schema for bootstrapping, but I do not fully understand exactly what comparisons were being made here. Specifically, what comparisons are contributing to the pairwise Z-scores? Does "pairwise comparison" refer to comparison of pairs of skeletal elements within a single species, or comparisons of the same measurement between pairs of species? Based on figure 2, I suspect it's the former, but this wasn't clear to me from the text.

I found the panels comprising figure 1 to generally be too small to be readable. I don't think the tree in panel A is useful, and so I recommend cutting it from this figure and moving to its own supplemental figure, where it can be shown much larger. Thus, the remaining space can be used to expand the remaining panels.

The microplots in panel A of figure 2 are far too small to be useful, and so I recommend they be moved to their own supplemental figure. Panel C is useful, but the text is far too small to read, so I recommend replacing the names of elements with numbers that are defined in the caption.

In figure 3, again, the taxonomic names are impossibly small. I don't find the tree useful, so I recommend cutting it, cutting the point labels in panels b-e, and adding a simple legend that connects point color to clade names.

RESPONSE TO REVIEWERS' COMMENTS

To the Reviewers:

I thank the reviewers for their insightful comments and recommendations. I have extended my analysis to a larger dataset of 228 bird species, in line with reviewer recommendations. The distribution of body masses in this larger sample is right skewed, because there is a large number of small birds represented in the dataset. I have therefore adapted method 2 and significance indices with a robust-resampling scheme. I have edited and improved all figures, increasing font sizes and adding taxon silhouettes to improve reader engagement.

A detailed list of changes and line indices is provided below: Responses are in bold and indicated with double chevrons: “>>”

Reviewer 1:

“Reviewer #1 (Remarks to the Author): This manuscript evaluates the link between body mass and the modularity (i.e., the structure of trait covariation (i.e., integration)) of size proportions across the skeleton in a rich sample of birds (although see comments below for improvement). The major finding of the study is an intriguing relationship between small body sizes and weaker size proportions integration in the wing skeleton which the authors explain as, mostly, reflecting the release of biomechanical constraints of flight associated with larger birds, allowing small birds to explore other optimal solutions to the construction of the wing (e.g., hyperaerial apodiiforms and hirundinids). The manuscript also proposes a novel methodological pipeline that opens interesting

avenues for future research. This manuscript excels in some ways such as the analytical component or the introduction to the topic of modularity and its generative factors at several evolutionary levels. However, I think there are several aspects that can be improved to enhance the accessibility of the findings overall. Furthermore, I have some additional concerns. First, I think it is important to clarify throughout the manuscript that when the authors are talking about integration and modularity this refers to relative sizes of skeletal elements. This must be clear from the title and the abstract. Personally, it took me a while to understand that the authors were not talking about shape/form integration.”

>> I have revised the title as follows:

“Birds of the tiny-verse: Avian miniaturisation is accompanied by a reorganisation of the modular evolution of skeletal proportions, with diverse implications across different bird groups,”

>> I have revised the abstract as follows:

Line 15: I now explicitly state “skeletal proportions”

Line 17: I now explicitly state “flight apparatus proportions”

*“Second, is there any reason why the reduced version of ‘TEMPO BIRDS’ is used instead of the largest and most updated version of this dataset (i.e., Navalón et al., 2022)? The expanded version includes ≈ 100 species many of which are passerines – a group including among the smallest birds and widely discussed in the present manuscript – but also some of the smallest representatives of other lineages (e.g., *Micropsitta*, *Psittaciformes*). Unless there is a major reason for not including this readily available dataset, I encourage the authors to make full use of the available data. If the reason – as I suspect – is the unavailability of flight style traits for the expanded dataset, then I reckon that: 1) this data mostly exists in large global datasets like AVONET”*

>> Reviewer 1 is correct that I selected Bjarnason et al., 2021’s dataset because I had already assembled complementary flight-style scores for all taxa. I agree with Reviewer 1 that this represents a minor portion of the analysis presented in the submitted manuscript, and that there are good reasons to use Navalón’s taxonomically expanded dataset.

>>I have re-run all core analyses with the dataset of Navalón et al., 2022. I thank Reviewer 1 for their suggestion. While I was undertaking this revision, I noticed that the taxon ‘*Uria*’ had been ascribed a body mass of 49.5 grams by Navalón; it should be closer to 1000 grams; I made this change during my analysis. This error had the potential to influence some analyses. I also noticed that the introduction of a large number of small birds from the full Navalón dataset has caused the distribution of log body mass to become right-skewed, which influenced the results of Breusch-Pagan tests (see method 2). This is because more intense sampling depresses the standard deviation of distributions under the central limits theorem. I mitigated this effect by augmenting method 2 with a sub-sampling strategy which draws many subsamples of taxa that are more likely to exhibit Gaussian distributions of log body mass. I compared results of these subsamples to a permuted dataset to assess whether Breusch-Pagan tests support heteroskedasticity/ a change in integration over body mass across birds.

>> I reviewed the available flight-style trait data on AVONET. It is categorical (rather than multivariate) and is not sufficiently granular for me to include in the current analyses. I would therefore need to score a further 79 bird flight-style descriptions under the expanded scheme of Taylor et al., 2014. Given that both reviewers identified this analysis as subsidiary to the core narrative of the paper, I have elected not to do this and rather concentrate on the central analyses of the manuscript. I think that the further exploration of the intersection between flight-style variety and the evolutionary organisation of skeletal proportions across birds is a subject for a future manuscript- especially because describing flight-style variety necessarily involves a semantic component.

“2) this latter part of the manuscript is such a minor aspect of the current study that it does not overweight the benefits of including a much richer dataset of smaller birds, and 3) links between body mass and species richness have been widely explored and probably (as admitted by the authors) are explained by many other factors related with generation times, metabolism, etc. Therefore, I recommend that even if the authors do not include the richer dataset that is readily available (which I encourage), excluding connections between flight style, richness and body mass will streamline an already (conceptually and analytically) complex paper.”

>> I agree with Reviewer 1 that the flight-style analysis is a minor aspect of the current work, and that it intersects with a much broader suite of explanatory variables.

“To be clear I think they could still discuss aspects of flight style biomechanics in the discussion which in reality they are not derived from their analyses.”

>> I agree with reviewer 1. The outcomes of the ecological analyses that I had performed in this work thus far have been ambiguous. I think that the representation of ecological variety in quantitative analyses is an especially difficult problem, because the perception of ecological difference is inherently a semantic decision on the part of the researcher. These are among the reasons I find questions about ecological disparification so interesting to explore, but I take Reviewer 1’s point that these questions are tangential to my submitted manuscript.

“More specific comments:

Introduction:

I can be wrong but I think the actual bird species count is closer to 11K than to 10K.”

>> Reviewer 1 is correct, I have amended the abstract at line 8 and I have added a citation to a recent review of avian biodiversity (Lee et al., 2022, state of the world’s birds [1]) at line 57, to reflect this. Thank you.

“Methods:

You need to state all the R functions and packages that you used.”

>>I have added references to all the R packages that I used in my methods sec-

tion, explicitly stated all functions used throughout analyses and provided version numbers. This information is also available as supporting comments in all codes deposited on my GitHub.

“Discussion:

A number of findings here resonate with the findings in Navalón et al., 2022. Including a clear environmental structure in patterns of morphological disparity across the skeleton – including but not restricted to skeletal proportions (e.g., much larger disparity of skeletal proportions in waterbirds with respect to landbirds)”

>> I have added the following commentary in my discussion at lines 179-184: “We repeatedly found that the birds which differed the most from general axes of trait covariation, especially involving trunk elements, belonged to subclade Aequirornithes, echoing the findings of Navalón et al. [2]. Our observation that these birds tend to be outliers in multiple directions, even though they are relatively closely related, is also commensurate with Navalón et al.’s conclusion that ecomorphological convergence is commonplace within water birds.”

>> I have added the following commentary in my discussion at lines 268-269: “Authors such as Navalón et al. have also suggested that non-passerine land-birds exploit the adaptation of peripheral skeletal elements to mediate ecological adaptation [2].”

>> I also observed in Navalón et al. 2022 that the avian wing is the only suite of skeletal proportions that is consistently less disparate than predicted by Brownian evolution, which is consistent with the avian wing evolving under strong adaptive constraint that limits the variety of forms it can explore. I have added the following commentary in my discussion at lines 270-273 to acknowledge this. “The wing is a notable outlier in our analysis, with no clear proximo-distal gradient. We suggest that strong evidence of the pervasive biomechanical constraint of this structure in fore-limb propelled birds (see Figure 4 Navalón et al. [2]) requires the consorted evolution of wing proportions as a single functional complex, tempering the proximo-distal gradient.”

“268 the highest values. The observation that more distal skeletal elements within the head and leg 269 tended to be more variable’ This result can be also interpreted as the effects of adaptation in the elements of the avian body that are in closest contact with the environment and therefore are more heavily shaped by selective pressures related to ecological traits as proven by an previous paper lead by the same first author (Orkney et al., 2021)”

>> I agree with Reviewer 1’s comment. I also note that Navalón comments that passerine-grade landbird disparification is accommodated disproportionately by peripheral elements such as the carpometacarpus.

Lines 263-268

>> “The disproportionate concentration of disparity within peripheral skeletal proportions could also reflect the fact that these are the elements interacting most directly with the substrate of the external environment- accommodating the manipu-

lation of food (mandible, rostrum and feet), arboreal, terrestrial, and aquatic locomotion (feet). These peripheral elements may therefore simply be more amenable to ecomorphological selection, as suggested by Orkney et al. [3]. ”

Line 270-276

>> “We suggest that strong evidence of the pervasive biomechanical constraint of this structure in fore-limb propelled birds (see Figure 4 Navalón et al. [2]) requires the consorted evolution of wing proportions as a single functional complex, tempering the proximo-distal gradient. This may explain why, although carpometacarpus size is strongly relevant to flight-style across birds, its 3D shape– which might not be evolving under the same constraints– exhibits an even clearer relationship to external wing geometry and flight-style variety [3,4].”

“I think the discussion could benefit from a summary statement that links back to the main evolutionary component of the study (i.e., links between body mass and the integration in skeletal size proportions)”

>> I agree, and I have added a paragraph that reflects and expands upon the content of the abstract, to create a symmetrical narrative across the manuscript:

Lines 307-321

>> “We conclude that our analysis of evolutionary covariances between the relative sizes of avian bones shows that the avian wing is a uniquely integrated structure within the avian body plan. We hypothesised that the strong consorted evolution of bones within the avian wing should be amplified under high body mass, which places tighter mechanical demands on this structure, and alleviated in birds with lower body mass. Our findings support this hypothesis, demonstrating that bones within the wing evolve more freely of each other in smaller birds. We also found that the evolution of wing bones becomes increasingly allied with the trunk in smaller birds. The evolutionary implications of this modular reorganisation are diverse. We suggest that weakened integration within the wing has facilitated the discovery of new wing arrangements and mechanical optima in apodiform birds, which have a disparate array of wing proportions (e.g. swifts, hummingbirds). By contrast, the evolution of skeletal proportions is more conservative in songbirds. We suggest that the coupling of wing and trunk evolution in this group facilitates their bounding flight-styles, which are restricted to small-bodied taxa, and that the ecological opportunities offered by passerine-style locomotion in cluttered environments may represent a key-innovation integral to their success in Cenozoic closed-canopy settings (e.g. [5]). ”

“Figures:

Figure 2c could be a different figure together with a version of Fig.2a with the inset heatmaps. Resulting Figure 2 would retain a,b,d,e and the schematic skeleton excluding the insets from 2a.”

>> I have extracted subplot c and the inset heatmaps of subplot a, and combined them into a new figure ‘Figure 3’.

“Figure 3. Some labels for clades and silhouettes to guide the reader’s attention. Same for

Figure 1. This will make the paper more appealing to a wider readership like the beautiful drawing of Figure 2.”

>> I have added labels for major avian subclades, silhouettes of a selection of representative taxa to the phylograms in both Figure 1 and Figure 4.

>> I have also changed the text size in subplots of Figure 4 to increase legibility and have added inset ‘flame-plots’ to illustrate significance of the depicted trends under a robust-resampling scheme (see methods 2).

Reviewer 2:

“Reviewer # 2 (Remarks to the Author):

“In this manuscript, Orkney and Hedrick use a dataset comprising morphometric data for a variety of skeletal elements collected from 149 volant bird species to investigate the evolutionary allometry of the functional modularity of the avian skeleton. The authors found that, in general, variability in skeletal element size increased with increasing body mass (especially more distal elements), as did strength of integration among the module comprising the wing elements; by contrast, strength of integration among the module comprising the trunk elements as well as between the wing and trunk modules decreased with body mass. Those patterns were supported by pairwise comparison of selected elements, revealing that increasing body mass was linked to tighter integration between humerus and carpometacarpus size, but weaker integration between the scapula and sternum and the carpometacarpus and sternum.

The patterns the authors seek to elucidate here are fundamental ones in bird evolution, and so this paper is likely to draw the interest of a wide range of functional, organismal and evolutionary biologists. The data collected and analyses used seem sufficient to investigate the allometric evolution of the sampled skeletal elements across birds, and the results a quite fascinating. I applaud the authors for making all the data and scripts that went into their analyses available – if only all authors were as forthcoming!

I have several concerns about the authors’ conclusions – in particular, about the assumptions made by the authors about the biological significance of their chosen data. There are also some parts of the paper I found quite confusing and unclear – specifically, the authors’ description of “method 1” and the discussion of “Pareto fronts.””

>> I have analysed a wider sample of birds from Navalón et al., 2022 (an additional 79 taxa), demonstrating the stability of our retrieved results providing persuasive evidence that our findings are general to crown birds.

>> I have re-written the description of method 1 in order to increase clarity.

>> I agree with Reviewer 2 about the concept of ‘Pareto’ fronts and have removed this content.

*“My first major concern regards the biological significance of the authors’ analyses. What the authors are specifically investigating is the (co)evolution of the allometry of the sizes of the skeletal elements sampled here. From there, the authors implicitly assume that there is a meaningful link between, on the one hand, evolutionary shifts in those allometries, and, on the other hand, functional shifts in some aspects of the modules comprising those elements. However, the authors do not explain why we might expect such a relationship, let alone what that relationship might be. To be clear, I am *not* suggesting that such a relationship does not exist – indeed, I find it quite intuitive that it would – but something approaching a clear, explicit understanding of that relationship must be provided by the authors to support any conclusions drawn by the analyses presented in this paper. I strongly urge the authors to establish this expected link in the introduction and explore what their results might tell us about that link in the discussion.”*

>> We thank Reviewer 2 for this insightful comment. This is an important point that needs to be clarified. We have added the following content throughout our introduction to accommodate this discussion, and make our predictions explicit for the reader:

>> 1) Lines 80-83

“We predict that smaller birds will experience lower mechanical stresses on the wings, and that the demand for evolutionary integration within their wings will therefore be lower than in their more massive relatives. Further, we predict that this may permit small-bodied bird lineages to explore a wider variety of wing proportions and functions.”

>> 2) Lines 62-65

Introduction paragraph 3 describes previous research in birds, explaining why they are a model group in which to explore this question. We have added the following text near the start of this paragraph, to explain the adaptive significance of modular re-organisation in avian lineages that have evolved smaller body sizes.

>> “However, the degree of evolutionary lability of modular organisation across birds– and its potential drivers– have yet to be systematically investigated, despite mosaic flexibility in the modular evolution of the avian locomotory system being an established explanation for avian evolvability [6, 7]. ”

>> 3) Lines 307-321

We have added a summarising paragraph at the end of the discussion which explicitly links back to the questions raised in the introduction. We identify hypotheses that explain these answers in Apodiformes and Passerines- and think that these would represent starting points for future research (especially sampling more densely within Strisores overall).

“Relatedly, when discussing the patterns they reconstruct, the authors must acknowledge the myriad other traits that might influence those patterns besides total size. Indeed, many other aspects of a given skeletal element are likely to influence the function of that element, arguably with greater strength – for example, degree of pneumatization, the relative development of sites for muscle attachment, or bone thickness and microstructure. Furthermore, many aspects of other,

related tissues are likely to major influence the evolution of skeletal modules – for example, associated musculature, and functionally related organs. This is especially relevant to the wing, to which the authors understandably devote much of their discussion. Indeed, one might easily imagine shifts in overall airfoil shape, feather distribution, and feather morphology profoundly influencing the function of the wing module without any appreciate shift in the sizes of the underlying skeletal elements.”

>> I focussed explicitly on skeletal proportions because it is known they exhibit a strong evolutionary modularity, which we hypothesise is either a reflection of fundamental evolutionary constraints or a functional regime that facilitates adaptation—perhaps both. I agree with Reviewer 2 that a variety of other biological variables are of great functional importance. I have therefore added the following discussion of these variables at Lines 185-199 in the Discussion:

>> “We reflect that a range of additional anatomical characteristics beyond skeletal proportions will also necessarily influence the biomechanical performance of wings and their resistance to stress under high loading. Possible examples include postcranial pneumatisation, which is known to differ characteristically between diving and non-diving birds [8]; wing musculature and feathers, which exhibit allometric scaling while wing bone proportions do not [9]; and external wing-shape, which has a complex relationship to foraging behaviour across birds exploiting aquatic resources [10]. Histological properties such as bone density and microstructure are also likely to have important roles in ecomorphological adaptation, for example the proportion of laminar bone varies between the wing skeletons of birds practicing fast-flapping and dynamic-soaring flight-styles [11], and there is an observed correlation between sites of strain and remodelled bone deposition in the furculae of soaring birds [12]. We advocate the exploration of this additional suite of variables, and the ways in which they may intersect with skeletal proportions to mediate functional adaptation and mitigate mechanical stresses, as subjects of future research. In addition, we reiterate that 3D skeletal shapes in birds, which can have greater ecological relevance than proportions at clade-wide scales (e.g. [3,4]) do not exhibit a clear modular organisation.”

“The authors do mention such potential influences in select cases, such as the sternum (but see my comment below), but these factors are largely ignored. I understand that testing for concomitant shifts in those other traits was beyond the scope of this study, but their exclusion must still temper the conclusions drawn by the authors from their analyses. I find it possible – perhaps likely – that quite different patterns might be recovered if other aspects of this system were sampled, instead. For example, I suspect that the shapes/morphologies of the trunk elements (e.g., the functional surfaces of the scapula and coracoid, the sizes of the keel and notches of the sternum) are far more directly linked to function than size, and thus might show a significantly different pattern of integration between the wing and trunk modules. The potential – indeed, likely – confounding intersection of these many traits should be discussed by the authors throughout their paper.”

>> I think Reviewer 2 is undoubtedly correct. Indeed, subtle shape evolution in the 3D form of avian bones does not have a clear modular structure, which stands in strong contrast to their skeletal proportions. I have made a note of this in the text

(Line 197-199) .”

>> I have also expanded the paragraph discussing the sternum in my discussion at lines 244-246: “We also reflect that previous work has demonstrated that 3D sternal shape adapts to a wide variety of ecological demands across birds [3, 13, 14], which may not be captured in a study that considers skeletal element sizes alone.”

“In many cases, the conclusions drawn by the authors are dependent on assumed evolutionary patterns that are not supported in this paper. The most crucial example is the pattern invoked in the title of the paper: miniaturization within birds. In many cases (most notably among passerines and hummingbirds), the authors explicitly assume that a lineage has been marked by miniaturization but provide no support for that assumption. In the introduction, the authors support the statement that birds have been “critically defined by miniaturization” (lines 63-64) by citing Benson et al. (2018), but that paper investigated trends in body mass leading to the stem lineage of extant birds, not among extant birds proper. If the authors were referring to papers cited within Benson et al. (2018), they should cite those papers, instead. If the authors mean to invoke their own analysis of body mass evolution, they should provide relevant discussion of those patterns. Nevertheless, the authors should provide support for any pattern of miniaturization they infer to justify the invocation of such a pattern in the title.”

>> I have added the following text to the introduction at Lines 69-78: “Indeed, ornithuromorphan birds, the stem that leads to crown birds, are distinguished as the only Mesozoic bird clade that flouts Cope’s rule [15]. A recent assessment of avian body masses across the K-Pg boundary implied that bird body mass evolution has followed diverse trajectories– with the Palaeognathae increasing in body mass, while Neoaves– which holds the majority of avian diversity– decreased in size towards the present day [16]. While ancestral state estimates of avian body mass remain challenging, it is likely that the hyper-diverse passerine subclade is distinguished by average body masses much smaller than those inferred in survivors of the K-Pg extinction, as well as most other avian subclades [17]. The rapidly-evolving hummingbird subclade [18], the earliest fossils of which emerge in the Oligocene, are also pervasively characterised by extremely small body size [19].”

>> I have also made it clear in the methods section that I imputed an ancestral reconstruction of Avian body mass, and specified in the results that the conclusion of this analysis suggests that the low masses of passerines, Apodiformes and Coraciimorphae are derived.

>> Line 363-365

“We imputed an ancestral state reconstruction of $\log_{10}(\text{mass})$ across our sample with the fastAnc() function of the *phytools* package, which we used to identify avian subclades that are typified by unusually small and derived body sizes.”

>> Line 103-105

“An ancestral state reconstruction (Figure 1a) indicates a moderate to large body mass (≈ 400 g) at the root of crown birds, and that the smaller body masses typical of Apodiformes, Passeriformes, and Coraciimorphae are therefore derived.”

“In other cases, the authors do support invoked patterns with citations, but they are not clear about what those patterns actually are or how they might be connected. For example, on lines 260-263, the authors compare their recovered pattern of increasing variability in skeletal proportions with increasing body size to a previously reported pattern of “high inherent variability of skeletal tissue.” As written, without going to the paper cited, I do not have a clue what “high inherent variability” means, and so I have no idea what the authors are trying to say here.”

>> I have expanded my discussion of this paper Lines 247-253:

“We found systemic increases in the variability of skeletal proportions among more massive birds (Figure 1b). This finding is compatible with previous observations by Hallgrímsson and Maiorana [20], which they attributed to the tendency for the contribution of skeletal tissue to organism mass to vary more than other tissue types, such as nervous tissue or viscera. Skeletal tissue tends to contribute a larger proportion of overall body composition in heavier animals, and this may contribute to greater variability at an interspecific scale in the allometry-corrected proportions of large birds’ skeletons. ”

“On lines 263-266, the authors propose a stronger functional link between the sizes of trunk elements and internal organs among smaller birds than larger birds without explaining why we might expect the organs to exert pressure on the sizes of those elements (especially the elements besides the sternum), and they do not provide a citation for sized-dependent variability in size of internal organs.”

>> Hallgrímsson and Maiorana’s work also suggests that viscera and nervous tissue mass is relatively less variable compared to skeletal and muscle tissue.

>> I have added the following text at Lines 247-258

“Additionally, we hypothesise that trunk skeletal elements in particular may conform more closely to the proportions of internal organs in small birds that may be approaching the physical limits of miniaturisation, which we suggest may temper the variability of trunk skeletal element sizes in comparison to larger bodied taxa. If correct, this hypothesis predicts stronger integration between soft tissue mass and trunk skeletal mass in smaller birds, both at an intra and interspecific scale.”

“On lines 242-244, the authors propose a functional link between trunk and wing integration and flight style specifically among passerines, but they do not explain what that link might be.”

>> Flap-bounding flight, in which a bird temporarily travels under ballistic motion between brief bursts of intense flapping, requires the wing to lie flush against the trunk to form a fusiform lift-generating surface. This should mean that, in flap-bounding birds, the wing and trunk dimensions are required to match and that they will evolve in unison.

>> Additionally we articulate this hypothesis at Lines 225-235:

“We suggest that the trunk and wing may be more integrated among small birds, and especially passerines, because they become unified as a single locomotory module in

intermittent and flap-bounding flight-styles, which are restricted to small birds. We suggest this requires the coupling of phenotypic variation in the trunk and wing, and note that the upper limit for flap-bounding flight is estimated at ≈ 300 g, [21, 22]. Birds larger than this mass tend to exhibit lower levels of within-trunk integration (Figure 2b) and birds with the most divergent carpometacarpus-sternum size ratios also tend to possess masses above 300 g (Figure 4e). It may therefore be possible that fundamental shifts in the flight-styles that are possible at different body masses underlie shifts in modular integration across birds, which suggests that—beyond determining evolutionary lability—evolutionary modularity may itself respond to adaptive demands in birds.”

“In lines 237-238, the authors suggest that the trunk does not “have an obvious locomotory function.” This is not true at all! The scapula, coracoid, and sternum are directly involved in the flight stroke!”

>> I agree and have removed this statement.

“I did not find any conclusions drawn from the authors’ analyses of species richness to be compelling, and I think that aspect can be cut from the manuscript without any cost to the utility of the study – which I urge the authors to do. A sample of only 149 species out of the more than 11,000 known species is sufficient to investigate the evolution of traits like skeletal element size because it can reasonably be assumed that one or few species are a reliable enough proxy for the condition characterizing the entire lineage to which those species belong. However, I am highly skeptical that the same can be said when species number is the very trait being investigated. Minimally, the authors should provide a metric of how reliably their sample serves as a proxy for total species richness and discuss potentially consequences of their low sample size, but again, I think all discussion of this can and should be removed.”

>> I agree and have removed this analysis.

“Similarly, I am not compelled by discussion of recovered patterns of ecological evolution beyond potential links to the wing. The authors’ chosen proxy for ecology is flight style, which we might expect to be mostly linked to the wing and potentially to the trunk, but only weakly linked to the skull and leg.”

>> I have removed the analysis of flight-style variety. The larger dataset of bird taxa that I am using does not yet have a suite of concurrent ecological scores, and it is no longer possible to conduct this analysis.

*“I found the discussion of “Pareto fronts” (paragraph 3 of the discussion) to be largely impenetrable. What I *think* they authors were trying to communicate is: “If the values of traits of interest are strongly linked among elements, then forces constraining the value of one such trait will also constrain the values of traits to which it is linked.” If my read there is incorrect, I urge the authors to “reverse engineer” how I might have come to such a misunderstanding. If my read is correct, it seems to me that this paragraph is largely redundant with the preceding paragraph, and so I recommend the authors fold whatever point they sought to make in this paragraph into that one. Minimally, the authors should reevaluate how they are approaching this point. Either way,*

I strongly urge the authors to remove direct discussion of “Pareto fronts” completely and to limit indirect discussion to a citation – it seems to me that, in order to clearly describe what they are and how they pertain, the authors will need to devote far too many column inches to the concept, distracting the reader from what we should be focused on.”

>> I have removed discussion of the concept of ‘Pareto fronts’; I agree with Reviewer 2 that this text did more to obfuscate than explain the results.

“Similarly, I found part of the description of “method 1” unclear. The authors did a fine job of outlining their treatment of body size bins and their schema for bootstrapping, but I do not fully understand exactly what comparisons were being made here. Specifically, what comparisons are contributing to the pairwise Z-scores? Does “pairwise comparison” refer to comparison of pairs of skeletal elements within a single species, or comparisons of the same measurement between pairs of species? Based on figure 2, I suspect it’s the former, but this wasn’t clear to me from the text.”

>> Reviewer 1 also requested clarification of method 1. I have re-written this section of the methods (Lines 306-401) to increase clarity. In essence, within each mass bin, I asked how likely pairs of bone sizes were likely to evolve together or independently. A stronger covariance resulted in a higher Z-score. Z-scores were computed for every possible pair of bones (n=78). ‘Pair-wise’ explicitly refers to pairs of bones. Analyses of their evolutionary covariance are inter-specific; conducted on many sub-samples of n=30 different species.

“I found the panels comprising figure 1 to generally be too small to be readable. I don’t think the tree in panel A is useful, and so I recommend cutting it from this figure and moving to its own supplemental figure, where it can be shown much larger. Thus, the remaining space can be used to expand the remaining panels.”

>> Reviewer 2’s recommendations differ from Reviewer 1’s, so I cannot make all the requested changes. I have removed taxon labels, because I agree this text was too small. I have added clade labels, which are larger and more legible, and which serve as a more useful guide to the structure of avian phylogeny than genus names can. I have expanded the remaining panels by removing the unnecessary legends and labelling the lines in subplots directly. I have also coloured these lines sympathetically with Figures 2 and 3, to increase aesthetic cohesion across the figures in this manuscript.

“The microplots in panel A of figure 2 are far too small to be useful, and so I recommend they be moved to their own supplemental figure. Panel C is useful, but the text is far too small to read, so I recommend replacing the names of elements with numbers that are defined in the caption.”

>> I have moved the microplots and subplot C to their own additional Figure (now Figure 3). The axes in this figure are larger and able to accommodate the element names.

“In figure 3, again, the taxonomic names are impossibly small. I don’t find the tree useful, so I recommend cutting it, cutting the point labels in panels b-e, and adding a simple legend that

connects point color to clade names.”

>> Based on recommendations from Reviewer 1, I have removed the taxon names and replaced them with clade labels and silhouettes which I think will better guide readers across avian phylogeny. I have coloured major avian subclades on these trees, and have matched the other subplot colour to them. I have reduced the point label number and increased their sizes to make them more legible. I think it is important to present the names of taxa that I mention in my discussion.

References

- [1] A. C. Lees, L. Haskell, T. Allinson, S. B. Bezeg, I. J. Burfield, L. M. Renjifo, K. V. Rosenberg, A. Viswanathan, and S. H. Butchart, “State of the world’s birds,” *Annual Review of Environment and Resources*, vol. 47, pp. 231–260, 2022.
- [2] G. Navalón, A. Bjarnason, E. Griffiths, and R. B. Benson, “Environmental signal in the evolutionary diversification of bird skeletons,” *Nature*, pp. 1–6, 2022.
- [3] A. Orkney, A. Bjarnason, B. C. Tronrud, and R. B. Benson, “Patterns of skeletal integration in birds reveal that adaptation of element shapes enables coordinated evolution between anatomical modules,” *Nature Ecology & Evolution*, vol. 5, no. 9, pp. 1250–1258, 2021.
- [4] T. L. Hieronymus, “Qualitative skeletal correlates of wing shape in extant birds (aves: Neoaves),” *BMC Evolutionary Biology*, vol. 15, no. 1, pp. 1–12, 2015.
- [5] S. L. Wing and B. H. Tiffney, “The reciprocal interaction of angiosperm evolution and tetrapod herbivory,” *Review of Palaeobotany and Palynology*, vol. 50, no. 1-2, pp. 179–210, 1987.
- [6] S. M. Gatesy and K. P. Dial, “Locomotor modules and the evolution of avian flight,” *Evolution*, vol. 50, no. 1, pp. 331–340, 1996.
- [7] K. P. Dial, “Evolution of avian locomotion: correlates of flight style, locomotor modules, nesting biology, body size, development, and the origin of flapping flight,” *The Auk*, vol. 120, no. 4, pp. 941–952, 2003.
- [8] M. G. P. Burton, R. B. Benson, and D. J. Field, “Direct quantification of skeletal pneumaticity illuminates ecological drivers of a key avian trait,” *Proceedings of the Royal Society B*, vol. 290, no. 1995, p. 20230160, 2023.
- [9] R. L. Nudds, G. W. Kaiser, and G. J. Dyke, “Scaling of avian primary feather length,” *PLoS One*, vol. 6, no. 2, p. e15665, 2011.
- [10] S. L. Baumgart, P. C. Sereno, and M. W. Westneat, “Wing shape in waterbirds: morphometric patterns associated with behavior, habitat, migration, and phylogenetic convergence,” *Integrative Organismal Biology*, vol. 3, no. 1, p. obab011, 2021.
- [11] E. L. Simons and P. M. O’connor, “Bone laminarity in the avian forelimb skeleton and its relationship to flight mode: testing functional interpretations,” *The Anatomical Record: Advances in Integrative Anatomy and Evolutionary Biology*, vol. 295, no. 3, pp. 386–396, 2012.

- [12] J. Mitchell, L. J. Legendre, C. Lefevre, and J. Cubo, “Bone histological correlates of soaring and high-frequency flapping flight in the furculae of birds,” *Zoology*, vol. 122, pp. 90–99, 2017.
- [13] T. M. Lowi-Merri, R. B. Benson, S. Claramunt, and D. C. Evans, “The relationship between sternum variation and mode of locomotion in birds,” *BMC biology*, vol. 19, no. 1, pp. 1–23, 2021.
- [14] T. M. Lowi-Merri, O. E. Demuth, J. Benito, D. J. Field, R. B. Benson, S. Claramunt, and D. C. Evans, “Reconstructing locomotor ecology of extinct avialans: a case study of ichthyornis comparing sternum morphology and skeletal proportions,” *Proceedings of the Royal Society B*, vol. 290, no. 1994, p. 20222020, 2023.
- [15] D. Hone, G. Dyke, M. Haden, and M. Benton, “Body size evolution in mesozoic birds,” *Journal of Evolutionary Biology*, vol. 21, no. 2, pp. 618–624, 2008.
- [16] C. R. Torres, M. A. Norell, and J. A. Clarke, “Bird neurocranial and body mass evolution across the end-cretaceous mass extinction: The avian brain shape left other dinosaurs behind,” *Science Advances*, vol. 7, no. 31, p. eabg7099, 2021.
- [17] N. E. Collias, “On the origin and evolution of nest building by passerine birds,” *The Condor*, vol. 99, no. 2, pp. 253–270, 1997.
- [18] J. A. McGuire, C. C. Witt, J. Remsen Jr, A. Corl, D. L. Rabosky, D. L. Altshuler, and R. Dudley, “Molecular phylogenetics and the diversification of hummingbirds,” *Current Biology*, vol. 24, no. 8, pp. 910–916, 2014.
- [19] G. Mayr, “Old world fossil record of modern-type hummingbirds,” *Science*, vol. 304, no. 5672, pp. 861–864, 2004.
- [20] B. Hallgrímsson and V. Maiorana, “Variability and size in mammals and birds,” *Biological Journal of the Linnean Society*, vol. 70, no. 4, pp. 571–595, 2000.
- [21] B. W. Tobalske, “Scaling of muscle composition, wing morphology, and intermittent flight behavior in woodpeckers,” *The Auk*, vol. 113, no. 1, pp. 151–177, 1996.
- [22] B. W. Tobalske, “Hovering and intermittent flight in birds,” *Bioinspiration & biomimetics*, vol. 5, no. 4, p. 045004, 2010.

REVIEWERS' COMMENTS

Reviewer #1 (Remarks to the Author):

I think the authors took on the comments and suggestions of both reviewers and addressed them carefully with additional analyses and thorough new discussion. I am very positively surprised by the attitude of the authors and I think this peer-review process has clearly improved the manuscript. I am very happy with the current state of this article.

I have however some minor observations about some parts of the article:

The authors wrote:

Indeed, ornithuromorph birds, the stem that leads to crown birds, are distinguished as the only Mesozoic bird clade that flouts Cope's rule [36].

This statement is unlikely to be true and I wouldn't rely on those conclusions because they are at odds with multiple lines of evidence:

1. A rebuttal to that article was published that exact year on the basis that the original article did not use phylogenetic methods – they reassessed the same data with PCMs and could not find any of the claimed trends.
2. Our knowledge of avian evolution has been boosted significantly since 2008.

I think the truth is that, at the moment, we cannot tackle this kind of studies using the Mesozoic fossil record. We need first to have a much richer and denser fossil record to be able to distinguish these sorts of trend. For an example of the kind of rich fossil record needed to do this see Sanisidro et al., 2022 in Science on body size evolution in brontotheres, a group of extinct odd-toed ungulates.

Instead, the authors may want to use this part of the Introduction to comment on how birds are the only group of dinosaurs that seemingly could evolve into smaller sizes that appear to have been 'forbidden' for the remainder of the dinosaur lineages (Benson et al., 2018). This could have implications for the lability of modularity in both groups – partly explaining this novel relationship between modularity and BM which may not characterise non-avian dinosaur evolution. The title is also too long in my opinion:

Birds of the tiny-verse: Avian miniaturisation is accompanied by a reorganisation of the modular evolution of skeletal proportions, with diverse implications across different bird groups

I suggest instead:

Birds of the tiny-verse: Avian miniaturisation is accompanied by a reorganisation of the modular evolution of skeletal proportions

But

Avian miniaturisation evokes (at least in the mind of palaeontologists) the events of miniaturisation leading to the origination of Avialae (total clade birds) as a clade from within non-avian dinosaurs. But the paper deals with the repeated evolution of miniaturisation within the crown group of birds – which is entirely different. I'd suggest the authors give some thought to this potential ambiguity and think about a way of mitigating it in the title. This will make the paper far more cited and avoid potential misunderstandings by the community. An example:

Birds of the tiny-verse: Repeated miniaturisation in crown birds was accompanied by a reorganisation of the modular evolution of skeletal proportions

Or

Birds of the tiny-verse: Small sizes in birds facilitated increased evolutionary lability of wing skeleton proportions

Figures:

Figure 1: Clade labels do not seem to cover all the taxa in the phylogeny – to avoid overlapping of labels I suggest intertwining light and dark grey bands – e.g., as per Cooney et al., 2017. This applies also to Figure 4.

Figure 3: To make part b more accessible I would add next to the heatmaps a silhouette of a bird whose size changes according to the body mass bins so readership can see at a glance the main message of the plots (i.e., changes in integration with size variation).

Typos:

Line 158: Spheniscus is misspelled

Reviewer #2 (Remarks to the Author):

I find the revised manuscript by Orkney and Hedrick to be much improved, and I commend the authors for their efforts. In particular, I find both the figures and the methods section to be clearer, and the additional discussion provided by the authors is effective. I have a few remaining larger – but not severe – concerns, and I have a number of suggestions.

My first concern again regards what is actually being investigated in this study: the allometric evolution of skeletal element size. I want to echo and emphasize a concern expressed by Reviewer 1 in their initial review: the authors should take care to clarify what is being investigated at all times. To the authors' credit, they have followed Reviewer 1's suggestion to increased clarity on this point in several places, but I think there are several instances where further clarity is warranted. For example, several topic sentences in the discussion (e.g., sentences beginning on lines 164, 200, 223, 260, and 270) make references to "modular organisation" or "integration" without clarifying what is being discussed. I suspect the authors are expecting that readers will infer from the larger text what is meant, but I find this to be confusing or – at worst – unintentionally misleading.

To this point, I again want to emphasize that any number of traits not tested here (e.g., shapes of the elements, shapes of associated structures like muscles or feathers) may explain the signals recovered in this paper. I appreciate that the authors added a paragraph to their discussion that explicitly discusses several of these potential factors, but they are largely ignored in discussions of patterns elsewhere in the paper. In particular, it is crucial to emphasize that patterns recovered here of low levels of variation in skeletal proportions (either within or between modules) may simply be offset (or even driven) by high levels of variation in (for example) element shape or associated integumentary structures.

I also think the authors may have misconstrued my initial point in my first review (lines 261-271 of their response document). I think the authors interpreted my comment as a call for more support for the concept of modularity across bird morphology – not at all what I meant! Rather, I was calling for stronger support of why we might expect absolute/relative skeletal size *specifically* to be modular, rather than some dimension(s) of shape. The authors do provide some justification for this hypothesis – increased body mass might be expected to correspond to increased loads on modules comprising the wing or leg – but little justification is given for why skeletal allometry – again, as opposed to some dimension(s) of shape – might be linked to other dimensions of life/organismal history, such as feeding ecology or reproductive mode. As in my previous review, I do not mean to suggest that I am skeptical such links might exist – again, I find it intuitive that they would – but the authors should make such hypotheses and their justifications explicit. I think addressing this point will help the authors address my previous point (i.e., more carefully interpreting patterns of low variability).

My second general concern again regards the invocation by the authors of "miniaturization" among birds. Specifically, I think the use of this concept unnecessarily limits the applicability of their results. Indeed, the body size-linked patterns recovered by the authors are significant regardless of whether a lineage has experienced episodes of miniaturization. By discussing those patterns largely in the context miniaturization, I think the authors undercut the power of those patterns by selectively applying them only to lineages for which miniaturization can be demonstrated, while simultaneously inviting criticism from folks like myself who might get distracted by potentially unjustified claims of miniaturization. By removing (or at least limiting) discussion of miniaturization, and instead focusing on the recovered patterns themselves, the authors will broaden the scope of potential impact of their study.

Relatedly, I appreciate that the authors have provided additional support for the notion that some lineages (e.g., passerines, apodiforms) have experienced episodes of miniaturization. However, if

the authors use their own results as part of this justification, those results should be discussed in the discussion in detail, rather than relegated to a single line in the results (i.e., lines 103-205). Moreover, the fact that typical body masses of a given clade (e.g., Apodiformes) are smaller than the body mass estimated for all of birds is not, on its own, sufficient to reconstruct a trend towards smaller masses. Rather, comparisons with near relatives and divergences – ideally in a quantitative framework – are necessary.

On a minor final note related to this point, the authors misrepresent the trend we recovered in Torres et al. (2021) (their citation number 37). In that study, we recovered a pattern of increasing body mass along the stem avian lineage leading towards to the divergence of the crown clade, followed by decrease towards the divergences of Neognathae and Neoaves; we do not directly comment on trends within Neoaves, including towards the present day – contra to lines 71-74.

Some smaller comments:

Regarding the title, I appreciate the nuance introduced by the authors, but I find the result to be a little clunky. And, again, I strongly advocate the authors not limit themselves to just “miniaturization.” A potential suggestion: “Birds of the tiny-verse: strength of avian skeletal proportion modularity scales with body size.”

I suggest rephrasing the hypotheses outlined in lines 80-83 so that the hypothesized pattern is described first, followed by the potential driver(s) of that pattern. I.e., “We hypothesize that integration of skeletal proportion within the elements comprising the wing will be weaker among smaller birds, driven in part by lower mechanical loads.” Thus, the sentence leads with the aspect actually being tested in this paper.

I remain uncompeled by the authors’ hypothesis that stronger trunk-wing integration within passerines is linked to flying style. Minimally, the authors should include their justification from their response document (i.e., lines 440-444 of the response document) in the main text. However, my doubts remain. First, I would expect the ability of birds to efficiently tuck their wings against their bodies to be far more strongly influenced by shape and range of motion of the wing, and to a lesser extent the shape of the body proper, than the size of the wing relative to the body. Second, many other clades of birds should theoretically be similarly optimized – most notably diving birds.

I question the authors’ phrasing in lines 45-47, where they describe “changes in skeletal proportions commensurate with body mass” as a “*driving force* of trait integration across the skeleton” (emphasis mine). Are those allometric changes really *driving* trait integration? Or has strength of integration in those traits been shown to *co-vary with* allometry? If a causal relationship really has been demonstrated, the authors should specify which traits are being driven.

A minor note, but I find “concentration of disparity” in line 263 to be paradoxical – maybe rephrase to, “The disproportionate disparity within...”

I approve of the compromises struck by the authors with regards to their figures, but I think the line weights in Fig. 1b got jumbled. In Fig. 4b, I find the colors of the data points almost impossible to discern because of the relatively heavy line weight of the black outlines – I suspect that line weight may already be at its minimum, so may be try dropping it altogether and seeing if the dots are more readable. If the authors insist on keeping labels in Fig. 4b, I urge them to place them more strategically to prevent overlapping with datapoints and – if possible – increasing their font size by at least one point.

A slight philosophical comment on lines 175-179 regarding expectations of modularity in ecologically-specialist avian subclades: in my opinion, definitions of “ecologically-specialist” are so necessarily narrow with regard to the amount of “biology” they describe (thereby failing to account for countless dimensions of life history/organismal biology by which they might be generalists, or by which other clades might be considered specialists in their own rights) that failures of patterns to match expectation are as likely to reflect biases in definitions as they are to reflect real biological signal.

Line 28 – need a space before second dash

Line 237 – need a space between “emphasise” and “the”

Reviewer #2 (Remarks on code availability):

Everything functions as expected.

RESPONSE TO REVIEWERS' COMMENTS

To the Reviewers:

We extend our thanks to the Reviewers and Editor for their continued support to help us develop and improve our manuscript.

A detailed list of changes and line indices is provided below: Responses are in bold and indicated with double chevrons: “>>”

Reviewer 1:

“I think the authors took on the comments and suggestions of both reviewers and addressed them carefully with additional analyses and thorough new discussion. I am very positively surprised by the attitude of the authors and I think this peer-review process has clearly improved the manuscript. I am very happy with the current state of this article. I have however some minor observations about some parts of the article:”

>> We thank Reviewer 1 for their continued help and support.

“The authors wrote:

Indeed, ornithuromorphan birds, the stem that leads to crown birds, are distinguished as the only Mesozoic bird clade that flouts Cope’s rule [36]. This statement is unlikely to be true and I wouldn’t rely on those conclusions because they are at odds with multiple lines of evidence:

1. A rebuttal to that article was published that exact year on the basis that the original article did not use phylogenetic methods – they reassessed the same data with PCMs and could not find any of the claimed trends.

2. *Our knowledge of avian evolution has been boosted significantly since 2008. I think the truth is that, at the moment, we cannot tackle this kind of studies using the Mesozoic fossil record. We need first to have a much richer and denser fossil record to be able to distinguish these sorts of trend. For an example of the kind of rich fossil record needed to do this see Sanisidro et al., 2022 in Science on body size evolution in brontotheres, a group of extinct odd-toed ungulates. Instead, the authors may want to use this part of the Introduction to comment on how birds are the only group of dinosaurs that seemingly could evolve into smaller sizes that appear to have been ‘forbidden’ for the remainder of the dinosaur lineages (Benson et al., 2018). This could have implications for the lability of modularity in both groups – partly explaining this novel relationship between modularity and BM which may not characterise non-avian dinosaur evolution.”*

>> **We have ammended the passage as follows: (Line 65)**
“**We also reflect that the evolutionary history of birds has been critically defined by the exploration of small sizes inaccessible to other dinosaur lineages, [1, 2], and that the smallest bodied avian subclades underwent rapid Cenozoic diversifications (e.g. [3]). This raises the possibility that there may be a novel intersection in birds between body mass and intrinsic controls on evolutionary dynamics such as modular organisation. ”**

“The title is also too long in my opinion: Birds of the tiny-verse: Avian miniaturisation is accompanied by a reorganisation of the modular evolution of skeletal proportions, with diverse implications across different bird groups I suggest instead: Birds of the tiny-verse: Avian miniaturisation is accompanied by a reorganisation of the modular evolution of skeletal proportions

But Avian miniaturisation evokes (at least in the mind of palaeontologists) the events of miniaturisation leading to the origination of Avialae (total clade birds) as a clade from within non-avian dinosaurs.

But the paper deals with the repeated evolution of miniaturisation within the crown group of birds – which is entirely different. I’d suggest the authors give some thought to this potential ambiguity and think about a way of mitigating it in the title. This will make the paper far more cited and avoid potential misunderstandings by the community. An example:

Birds of the tiny-verse: Repeated miniaturisation in crown birds was accompanied by a reorganisation of the modular evolution of skeletal proportions Or Birds of the tiny-verse: Small sizes in birds facilitated increased evolutionary lability of wing skeleton proportions,”

>> **We have ammended the title as follows: (Line 1)**
“**Small body size facilitates increased evolutionary lability of wing skeleton proportions in birds,”** We made a minor alteration of the title suggested by Reviewer 1 so that we include the phrase ‘body size’ in the title.

“Figures:

Figure 1: Clade labels do not seem to cover all the taxa in the phylogeny – to avoid overlapping of labels I suggest intertwining light and dark grey bands – e.g., as per Cooney et al., 2017. This applies also to Figure 4.”

>> **I have intertwined black and dark grey bands on the phylogram, to resemble Cooney et al., 2017’s aesthetic. I have also added ticks to the groups representing the hoatzin and owls, so that they are more clearly identified. I have made this change**

to Figure 1 and to Figure 4. In addition, Reviewer 2 requested that I remove the line weight on points in Figure 4, increase the label point size and alter label positions.

“Figure 3: To make part b more accessible I would add next to the heatmaps a silhouette of a bird whose size changes according to the body mass bins so readership can see at a glance the main message of the plots (i.e., changes in integration with size variation).”

>> I have added a silhouette of the thrush, *Turdus pilaris*, scaled according to the log₁₀(mass) associated with each subplot in panel b.

“Typos:

Line 158: Spheniscus is misspelled”

>> I have addressed this spelling error.

Reviewer 2:

“Reviewer # 2 (Remarks to the Author):

I find the revised manuscript by Orkney and Hedrick to be much improved, and I commend the authors for their efforts. In particular, I find both the figures and the methods section to be clearer, and the additional discussion provided by the authors is effective. I have a few remaining larger – but not severe – concerns, and I have a number of suggestions. My first concern again regards what is actually being investigated in this study: the allometric evolution of skeletal element size. I want to echo and emphasize a concern expressed by Reviewer 1 in their initial review: the authors should take care to clarify what is being investigated at all times. To the authors’ credit, they have followed Reviewer 1’s suggestion to increased clarity on this point in several places, but I think there are several instances where further clarity is warranted. For example, several topic sentences in the discussion (e.g., sentences beginning on lines 164, 200, 223, 260, and 270) make references to “modular organisation” or “integration” without clarifying what is being discussed. I suspect the authors are expecting that readers will infer from the larger text what is meant, but I find this to be confusing or – at worst – unintentionally misleading. To this point, I again want to emphasize that any number of traits not tested here (e.g., shapes of the elements, shapes of associated structures like muscles or feathers) may explain the signals recovered in this paper. I appreciate that the authors added a paragraph to their discussion that explicitly discusses several of these potential factors, but they are largely ignored in discussions of patterns elsewhere in the paper. In particular, it is crucial to emphasize that patterns recovered here of low levels of variation in skeletal proportions (either within or between modules) may simply be offset (or even driven) by high levels of variation in (for example) element shape or associated integumentary structures.”

>> We have identified the instances on lines 164, 200, 223, 260, 270 and added clarifying language to ensure that readers know that the modular organisation of skeletal proportions is the subject of our work.

>> Line 164 (now line 160)

has been revised as “The application of our tandem-method approach reveals an intersection between body mass evolution and the modular organisation of skeletal

proportions that coheres with biomechanical expectations.” We have chosen the language ‘modular organisation of skeletal proportions’ for brevity.

>> Line 200 (now line 206)
has been revised as “We hypothesise that higher integration of skeletal proportions within the wing reflects the influence of tighter mechanical constraints at high body masses, reasoning that changes in the mechanical properties of one trait within a highly-constrained structure will require compensatory changes in other traits to maintain function.”

>> Line 223 (now line 229)
has been revised as “We suggest that songbirds play a substantial role establishing strong evolutionary integration between the skeletal proportions of the wing and trunk in small-bodied birds.”

>> Line 260 (now line 272)
has been revised as “The observation that the relative sizes of more distal skeletal elements within the head and leg tend to be more variable (Figure 1b; Head, Leg) is consistent with predictions under a proximo-distal gradient of embryogenesis [4] and previous observations of greater evolutionary variance in more rostral components of the avian cranium [5].”

>> Line 270 (now line 283)
has been revised as “The wing is a notable outlier in our analysis, with no clear proximo-distal gradient in the variability of the relative size of different skeletal elements.”

>> Line 65 (now line 60)
has been revised as “However, the degree of evolutionary lability of modular organisation across birds– and its potential drivers– have yet to be systematically investigated, despite mosaic flexibility in the modular evolution of the proportions of the the avian locomotory system being an established explanation for avian evolvability [6,7].”

>> Line 80 (now line NA) has been removed during revision.

>> Line 84 (now line 85)
has been revised as “Here, we develop a novel framework to untangle the complexity inherent to modularity and explore the impact of body mass on the modular organisation of skeletal proportion evolution in birds.”

>> Line 105 (now line 110)
has been revised as “The relative sizes of the more distal skeletal elements within the head and leg (e.g. mandible, tarsometatarsus) are the most variable. However, there is no clear evidence that the relative sizes of more distal elements within the wing are more variable than proximal elements. Furthermore, the relative sizes of skeletal elements within the trunk exhibit a disorganised increase in variance with body mass. We note that those skeletal elements which did not conform to our expectations are

generally major components of the avian flight apparatus, a point we return to in the discussion.”

>> Line 215 (now line 221)

has been revised as “The leg and head hence experience no trade-off requiring the increased consolidation of elements proportions to resist greater mechanical stress at the expense of evolvability in more massive birds. ”

>> Line 225 (now line 232)

has been revised as “We suggest that the relative sizes of bones within the trunk and wing may be more integrated among small birds, and especially passerines, because they become unified as a single locomotory module in intermittent and flap-bounding flight-styles, which are restricted to small birds.”

>> Line 283 (now line 297)

has been revised as “Additional support is lent to this hypothesis by our finding that the relative size of the scapula, which has a direct role in the avian flight apparatus, is not characterised by greater levels of variance compared with other trunk elements, despite occupying a more peripheral position within the trunk.”

I have also clarified my description of other studies:

>> Line 22 has been revised as

“For example, morphological variation in mammalian vertebrae clusters into functional and developmental regions along the spinal column [8,9] and morphological variety of avian cranial bones exhibit stronger evolutionary correlations within tissues of the cranial neural crest and mesodermal lineages [5]”

>> In response to Reviewer 2’s discussion of a broader diversity of possible traits, such as 3D shape, musculature and plumage: It has previously been shown that skeletal proportions exhibit a much stronger signal of modular evolution than 3D skeletal shape in birds. Essential changes in skeletal proportion are, in the absence of situations such as mechanical equivalence, likely to change the essential mechanical function of the skeleton, with substantial expected implications for activities such as flight mechanics. 3D shape, by contrast can be subtly modified in many ways that can achieve slight changes to essential mechanical function or other ancillary functions (for example, the alteration of a tarsometatarsal bone shape to accommodate raptorial dissection of prey items). It is not completely clear how these changes in 3D shape, occurring independently of skeletal proportions, are likely to affect the modular integration of skeletal proportions as body mass changes, but I agree with Reviewer 2 that an acknowledgement of their potential effects and a description of their expected influence on our results should be described from the outset of the manuscript.

We therefore entertain the possibility that the remiges, which are more flexible than bone, may be able to occupy a larger section of the distal aerofoil surface at the expense of the carpometacarpus in small birds (indeed, the relative remige length as a proportion of the wing scales allometrically). Assuming that the avian wing’s

gross proportions are isometric with respect to body mass, variation in remige length could result in greater evolutionary variability of the carpometacarpus bone length, relative to other wing bones, and hence a decrease in perceived integration of skeletal proportions. In this hypothetical scenario, the accessory mechanical role of ancillary structures, such as feathers, are able to accommodate a portion of the consolidating structural role that might be more incumbent on the skeleton in massive birds. We are therefore going to include words to this affect at the penultimate paragraph of our introduction:

>> Line 75 (now line 74)

“We observe that the primary feathers occupy a greater portion of the total avian wingspan in small birds [10], while individual skeletal proportions scale with isometry [11]. We hypothesise that the accessory mechanical function of ancillary structures with lower stiffness than the wing skeleton, such as feather remiges and supporting musculature, is able to accommodate a portion of the consolidating structural role that might be more incumbent on the skeleton in massive birds experiencing greater mechanical stresses. We therefore have a clear reason to anticipate weaker integration of the wing skeleton proportions within small birds. Furthermore, [10] suggests that longer primary remiges facilitate differential joint placement along the avian wing and access to novel wing stroke kinematics, leading us to expect that a reduction in integration within the avian wing skeleton may permit the exploration of a broader remit of flight-styles. ”

This new paragraph required the removal of a single adjacent line that would have become redundant.

As an aside, we speculate the process by which the flight feathers take up a larger portion of the wing in some groups of birds- moving the skeletal joints proximally- could be viewed as an aerial analogue to the evolution of digitigrade and unguligrade postures in terrestrial vertebrates.

>> Line 72, (now line 73)

I have changed ‘volant’ to ‘forelimb-propelled,’ for two reasons:

- 1) There is a single penguin in the dataset
- 2) A large number of other researchers I have relayed my research to were unfamiliar with the word ‘volant’

>> Line 90 (now line 93)

I am now concluding the introduction with a specific call for other researchers to pursue the questions identified by Reviewer 2:

“In particular, we advocate the further exploration of 3D skeletal morphology, remige proportions and soft tissue anatomy in bird wings.”

I would greatly like to explore these questions myself- I note that Nudds et al., 2011’s sample includes only 34 bird species, and that specific measurements of individual bone lengths in the wing can be complicated in study skins. I have observed remiges occasionally in uCT scan data before, and I wonder whether data on MorphoSource could be marshalled to this aim across a large dataset of birds- potentially all the

birds in Navalon et al., 2022 will be available on Morphosource and their specific remige lengths could be collected under the auspices of a new project. The further exploration of soft tissue anatomy such as musculature is a subject that I think would require dissection of recently dead birds that have not experienced shrinkage, drying or immersion in preservatives. Perhaps it can be shown that muscle attachment footprints differ characteristically across birds as a function of body mass, or that the tensile strength of muscle- relative to resistance to flexure of bones- is greater in small birds (this sounds like it should be true, if the intensive properties of muscle and bone are relatively consistent across birds, but it would be very interesting to be proven wrong).

*“I also think the authors may have misconstrued my initial point in my first review (lines 261-271 of their response document). I think the authors interpreted my comment as a call for more support for the concept of modularity across bird morphology – not at all what I meant! Rather, I was calling for stronger support of why we might expect absolute/relative skeletal size *specifically* to be modular, rather than some dimension(s) of shape. The authors do provide some justification for this hypothesis – increased body mass might be expected to correspond to increased loads on modules comprising the wing or leg – but little justification is given for why skeletal allometry – again, as opposed to some dimension(s) of shape – might be linked to other dimensions of life/organismal history, such as feeding ecology or reproductive mode. As in my previous review, I do not mean to suggest that I am skeptical such links might exist – again, I find it intuitive that they would – but the authors should make such hypotheses and their justifications explicit. I think addressing this point will help the authors address my previous point (i.e., more carefully interpreting patterns of low variability).”*

>> We have added more specific language in our introduction, to justify our predictions, (see lines 73 onwards), and have also related these justifications to potential futurework investigating soft tissue structures. We have cited Nudds 2011 to justify the hypothesis that variation in the allometry of wing bone proportions should correspond to variety in flight kinematics and therefore ecological variables such as flight styles.

“My second general concern again regards the invocation by the authors of “miniaturization” among birds. Specifically, I think the use of this concept unnecessarily limits the applicability of their results. Indeed, the body size-linked patterns recovered by the authors are significant regardless of whether a lineage has experienced episodes of miniaturization. By discussing those patterns largely in the context miniaturization, I think the authors undercut the power of those patterns by selectively applying them only to lineages for which miniaturization can be demonstrated, while simultaneously inviting criticism from folks like myself who might get distracted by potentially unjustified claims of miniaturization. By removing (or at least limiting) discussion of miniaturization, and instead focusing on the recovered patterns themselves, the authors will broaden the scope of potential impact of their study.”

>> We agree with Reviewer 2 that using the term ‘miniaturisation’ could limit the potential impact of our study, especially given that it may be confused with a reduction in body mass in stem birds. I have therefore carefully removed reference to ‘miniaturisation’ throughout the manuscript and instead used language such as ‘size

scaling'. Because these instances are numerous I have not provided specific line numbers in my response. We have amended this language in the abstract, introduction, I have amended the language that low body masses within apodiforms, passerines and coraciiforms are 'therefore derived' to 'likely to be derived' (line 110) I have amended language in the opening paragraph of the discussion to remove the word 'miniaturise'.

"Relatedly, I appreciate that the authors have provided additional support for the notion that some lineages (e.g., passerines, apodiforms) have experienced episodes of miniaturization. However, if the authors use their own results as part of this justification, those results should be discussed in the discussion in detail, rather than relegated to a single line in the results (i.e., lines 103-205). Moreover, the fact that typical body masses of a given clade (e.g., Apodiformes) are smaller than the body mass estimated for all of birds is not, on its own, sufficient to reconstruct a trend towards smaller masses. Rather, comparisons with near relatives and divergences – ideally in a quantitative framework – are necessary."

>> I agree with Reviewer 2. I do not necessarily believe that the history of body mass evolution across birds can be conclusively reconstructed from extant birds alone, and therefore- while I view it as likely that the lineages leading to songbirds and hummingbirds underwent reductions in body size, I have made sure that the language of the manuscript does not present this as an accepted truth and that claims within the manuscript are not contingent upon this hypothesis.

"On a minor final note related to this point, the authors misrepresent the trend we recovered in Torres et al. (2021) (their citation number 37). In that study, we recovered a pattern of increasing body mass along the stem avian lineage leading towards to the divergence of the crown clade, followed by decrease towards the divergences of Neognathae and Neoaves; we do not directly comment on trends within Neoaves, including towards the present day – contra to lines 71-74."

>> I have removed the content relevant to this mistake in lines 71-74. Thankyou for pointing this out.

I have revised this content as: (Line 256)

"We also reflect that the evolutionary history of birds has been critically defined by the exploration of small sizes inaccessible to other dinosaur lineages, [1,2], and that the smallest bodied avian subclades underwent rapid Cenozoic diversifications (e.g. [3]), although the history of body size evolution through the Mesozoic remains fragmentary [12]."

Some smaller comments:

Regarding the title, I appreciate the nuance introduced by the authors, but I find the result to be a little chunky. And, again, I strongly advocate the authors not limit themselves to just "miniaturization." A potential suggestion: "Birds of the tiny-verse: strength of avian skeletal proportion modularity scales with body size."

>>We have rephrased the title as "Small body size facilitates increased evolutionary lability of wing skeleton proportions in birds" following a compromise of the

advice provided to us by both reviewers and the journal.

I suggest rephrasing the hypotheses outlined in lines 80-83 so that the hypothesized pattern is described first, followed by the potential driver(s) of that pattern. I.e., “We hypothesize that integration of skeletal proportion within the elements comprising the wing will be weaker among smaller birds, driven in part by lower mechanical loads.” Thus, the sentence leads with the aspect actually being tested in this paper.

>> We have rephrased the paragraph as follows: (Lines 76) “We hypothesise that the accessory mechanical function of ancillary structures with lower stiffness than the wing skeleton, such as feather remiges and supporting musculature, are able to accommodate a portion of the consolidating structural role that might be more incumbent on the skeleton in massive birds experiencing greater mechanical stresses. We therefore have a clear reason to anticipate weaker integration of wing skeleton proportions within small birds. Furthermore, Nudds et al. [10] suggest that longer primary remiges facilitate differential joint placement along the avian wing and access to novel wing stroke kinematics in smaller birds, leading us to expect that a reduction in integration within the avian wing skeleton may permit the exploration of a broader remit of flight-styles.”

This mechanical justification incorporates hypotheses about the potential roles of musculature and plumage which the Reviewers requested. I acknowledge that the hypothesised pattern that results from this mechanism is described as a corollary, rather than the leading clause of the sentence. I believe this style is necessary here so that the reader feels that they can arrive at this hypothesis as a deduction drawn from the presented observations.

“I remain uncompelled by the authors’ hypothesis that stronger trunk-wing integration within passerines is linked to flying style. Minimally, the authors should include their justification from their response document (i.e., lines 440-444 of the response document) in the main text. However, my doubts remain. First, I would expect the ability of birds to efficiently tuck their wings against their bodies to be far more strongly influenced by shape and range of motion of the wing, and to a lesser extent the shape of the body proper, than the size of the wing relative to the body. Second, many other clades of birds should theoretically be similarly optimized – most notably diving birds.”

>> We have added the following text at Line 235
“Flap-bounding flight, in which a bird temporarily travels under ballistic motion between brief bursts of intense flapping [13], requires the wing to lie flush against the trunk to form a fusiform lift-generating surface. This could mean that, in flap-bounding birds, the wing and trunk dimensions are required to match and that their capacity to evolve independently is diminished compared to other birds.”

I note that many diving birds do not tuck their wing flush against their body during dives; gannets do not for example; their wings are so long that they are swept back to trail the body before breaking the water surface and are subsequently deployed as fins. Little auks, guillemots frigate birds, puffins and penguins all have various wing postures during dives, which cannot be described as ‘flush’. I note with interest that dippers, which are passerine birds we might expect to practice intermittent flight,

do not fold their wings flush during dives and that they practice high frequency continuous flapping flight. Amazingly, kingfishers practice continuous flapping hovering flight. Kingfishers may break the water surface during dives with their wings trailing behind them like a gannet or even with their wings open, as though the transition in fluid medium is an irrelevance that they hardly notice. There is perhaps a greater diversity of tucking styles in larger birds as well, such as the swan, which tucks its wings over its back. I agree with Reviewer 2 that the flap-bounding hypothesis could be challenged- and would welcome a response from other researchers if they believe they have falsified it.

*“I question the authors’ phrasing in lines 45-47, where they describe changes in skeletal proportions commensurate with body mass” as a “*driving force * of trait integration across the skeleton” (emphasis mine). Are those allometric changes really *driving* trait integration? Or has strength of integration in those traits been shown to *co-vary with* allometry? If a causal relationship really has been demonstrated, the authors should specify which traits are being driven.”*

>> We agree with Reviewer 2’s assessment. We have changed the language ‘driving’ to ‘covary’.

A minor note, but I find “concentration of disparity” in line 263 to be paradoxical – maybe rephrase to, “The disproportionate disparity within...”

>> We have rephrased this as (Line 275)
“The relatively high phenotypic disparity of peripheral skeletal proportions, compared to the proximal skeleton,”

“I approve of the compromises struck by the authors with regards to their figures, but I think the line weights in Fig. 1b got jumbled. In Fig. 4b, I find the colors of the data points almost impossible to discern because of the relatively heavy line weight of the black outlines – I suspect that line weight may already be at its minimum, so may be try dropping it altogether and seeing if the dots are more readable. If the authors insist on keeping labels in Fig. 4b, I urge them to place them more strategically to prevent overlapping with datapoints and – if possible – increasing their font size by at least one point.”

>> We have reduced the linewidth in Figure 1b to increase clarity. We have removed the black border around points in Figure 4b. We have reduced the number of taxon labels in Figure 4b, and have increased their size as much as the figure plan allows (Nature journals require text in figures to be between 5-7 pt), taking effort to place labels more strategically.

“A slight philosophical comment on lines 175-179 regarding expectations of modularity in ecologically-specialist avian subclades: in my opinion, definitions of “ecologically-specialist” are so necessarily narrow with regard to the amount of “biology” they describe (thereby failing to account for countless dimensions of life history/organismal biology by which they might be generalists, or by which other clades might be considered specialists in their own rights) that failures of patterns to match expectation are as likely to reflect biases in definitions as they are to reflect real biological signal.”

>> We have reworded this sentence as follows: (Line 181)

“The structure of modular aggregation and dis-aggregation we have uncovered here is disseminated across avian phylogeny (Figure 4), which defies expectations that the emphasis of distinct combinations of modularity within a broader avian ‘locomotor mosaic’ should correspond strongly with avian subclades distinguished by unique combinations of ecology, body size and developmental mode (e.g. Dial, 2003 Figure 3 [7]).”

“Line 28 – need a space before second dash”

>> I have added a space

Line 237 – need a space between “emphasise” and “the” ”

>> I have added a space

Reviewer #2 (Remarks on code availability):

“Everything functions as expected.”

>> Thank you for independently assessing the code. I wish to note that I realised the ribbons on Figures 2 and 4 were not tracing the exact 1 and 2 σ quantiles. I have fixed this, and it is an aesthetic-only change, but I am declaring it for full transparency.

References

- [1] G. Mayr, “Old world fossil record of modern-type hummingbirds,” *Science*, vol. 304, no. 5672, pp. 861–864, 2004.
- [2] R. B. Benson, G. Hunt, M. T. Carrano, and N. Campione, “Cope’s rule and the adaptive landscape of dinosaur body size evolution,” *Palaeontology*, vol. 61, no. 1, pp. 13–48, 2018.
- [3] J. A. McGuire, C. C. Witt, J. Remsen Jr, A. Corl, D. L. Rabosky, D. L. Altshuler, and R. Dudley, “Molecular phylogenetics and the diversification of hummingbirds,” *Current Biology*, vol. 24, no. 8, pp. 910–916, 2014.
- [4] B. Hallgrímsson, K. Willmore, and B. K. Hall, “Canalization, developmental stability, and morphological integration in primate limbs,” *American Journal of Physical Anthropology: The Official Publication of the American Association of Physical Anthropologists*, vol. 119, no. S35, pp. 131–158, 2002.
- [5] R. N. Felice and A. Goswami, “Developmental origins of mosaic evolution in the avian cranium,” *Proceedings of the National Academy of Sciences*, vol. 115, no. 3, pp. 555–560, 2018.
- [6] S. M. Gatesy and K. P. Dial, “Locomotor modules and the evolution of avian flight,” *Evolution*, vol. 50, no. 1, pp. 331–340, 1996.

- [7] K. P. Dial, “Evolution of avian locomotion: correlates of flight style, locomotor modules, nesting biology, body size, development, and the origin of flapping flight,” *The Auk*, vol. 120, no. 4, pp. 941–952, 2003.
- [8] K. E. Jones, K. D. Angielczyk, and S. E. Pierce, “Stepwise shifts underlie evolutionary trends in morphological complexity of the mammalian vertebral column,” *Nature communications*, vol. 10, no. 1, pp. 1–13, 2019.
- [9] K. E. Criswell, L. E. Roberts, E. T. Koo, J. J. Head, and J. A. Gillis, “hox gene expression predicts tetrapod-like axial regionalization in the skate, *leucoraja erinacea*,” *Proceedings of the National Academy of Sciences*, vol. 118, no. 51, p. e2114563118, 2021.
- [10] R. L. Nudds, G. W. Kaiser, and G. J. Dyke, “Scaling of avian primary feather length,” *PLoS One*, vol. 6, no. 2, p. e15665, 2011.
- [11] R. L. Nudds, “Wing-bone length allometry in birds,” *Journal of Avian Biology*, vol. 38, no. 4, pp. 515–519, 2007.
- [12] C. R. Torres, M. A. Norell, and J. A. Clarke, “Bird neurocranial and body mass evolution across the end-cretaceous mass extinction: The avian brain shape left other dinosaurs behind,” *Science Advances*, vol. 7, no. 31, p. eabg7099, 2021.
- [13] B. W. Tobalske, “Hovering and intermittent flight in birds,” *Bioinspiration & biomimetics*, vol. 5, no. 4, p. 045004, 2010.